# Differential responses of the gut microbiome and resistome to antibiotic exposures in infants and adults

Xuanji Li [1], Asker Brejnrod [2], Jonathan Thorsen [3], Trine Zachariasen[2], Urvish Trivedi[1], Jakob Russel[1], Gisle Alberg Vestergaard [2], Jakob Stokholm [3,4], Morten Arendt Rasmussen [3,4] ✉ & Søren Johannes Sørensen [1] ✉

Despite their crucial importance for human health, there is still relatively limited knowledge on how the gut resistome changes or responds to antibiotic treatment across ages, especially in the latter case. Here, we use fecal meta-genomic data from 662 Danish infants and 217 young adults to fill this gap. The gut resistomes are characterized by a bimodal distribution driven by *E. coli* composition. The typical profile of the gut resistome differs significantly between adults and infants, with the latter distinguished by higher gene and plasmid abundances. However, the predominant antibiotic resistance genes (ARGs) are the same. Antibiotic treatment reduces bacterial diversity and increased ARG and plasmid abundances in both cohorts, especially core ARGs. The effects of antibiotic treatments on the gut microbiome last longer in adults than in infants, and different antibiotics are associated with distinct impacts. Overall, this study broadens our current understanding of gut resistome dynamics and the impact of antibiotic treatment across age groups.

The rampant use of antibiotics has escalated the spread of antibiotic resistance among bacteria to the point where multi-drug resistant infections have become untreatable, posing a major challenge for modern medicine[1,2]. The indigenous bacteria residing in the human gut[3] constitute a large reservoir of antibiotic resistance genes (ARGs) which they exchange among themselves and with pathogens through horizontal gene transfer[4,5]. A comprehensive understanding of antibiotic resistance profiles and the ARG-carrying bacteria in the human gut is essential for developing novel intervention strategies to minimize the spread of antibiotic resistance. Metagenomic sequencing has provided initial characterizations of ARGs in the human gut microbiome[6–8], yet the links between antibiotic use, age, bacterial hosts, and ARGs remain poorly explored, particularly in large human cohorts.

Antibiotic resistance emerges in the infant gut through early colonization by bacteria, mainly acquired from the mother[9,10] and environmental exposures[11–13]. Previous work by our group described how the infant gut serves as a reservoir of ARGs, with *E. coli* being the largest single contributor[14]. Through the first years of life, the gut microbiome gradually comes to resemble that of adults, after which it is believed to be relatively stable[15]. Many studies have shown a higher level of gut ARGs in infants than in adults[9,16], but extensive investigations into ARG profiles between them are relatively rare. However, this information is necessary to understand the spread and succession of ARGs and to improve antibiotic stewardship in infants and adults.

More generally, the problem of antibiotic resistance can only be addressed through an improved understanding of the effects of antibiotics on the host microbiome, and how these might differ at different ages and life stages. It is well known that antibiotic treatments can have negative effects on the gut microbiome[17–19]. Given the differences in community composition, stability, and resilience between infant and adult gut assemblages, the manner and extent

[1]Department of Biology, Section of Microbiology, University of Copenhagen, 2100 Copenhagen, Denmark. [2]Department of Health Technology, Technical University of Denmark, Section of Bioinformatics, 2800 Kgs Lyngby, Denmark. [3]COPSAC, Copenhagen Prospective Studies on Asthma in Childhood, Herlev and Gentofte Hospital, University of Copenhagen, Copenhagen, Denmark. [4]Department of Food Science, Section of Microbiology and Fermentation, University of Copenhagen, 1958 Frederiksberg C, Denmark. ✉e-mail: mortenr@food.ku.dk; sjs@bio.ku.dk

to which the microbiome responds and recovers from antibiotic treatment may vary with age. For example, an animal study showed that the recovery of gut microbes from antibiotic treatment was affected by host diet, bacterial community structure, and host living environment[20]. However, with respect to differences between the infant and adult gut microbiome in humans, the response variance to conventional antibiotic therapy has not been fully explored, although such information is critical for understanding how antibiotics remodel the gut.

In this cross-sectional sub-study of the COPSAC cohort, we sequenced fecal metagenomes from COPSAC cohort of 217 young adults (Table 1), aged 18 years (median age), and used metagenomic bins to associate ARGs with their bacterial hosts, thus gaining insight into the distribution of ARGs across bacterial species. Moreover, we comprehensively compared the abundance and community composition of ARGs (in bacteria as well as plasmids) and ARG-carrying hosts between these adults and 662 1-year-old (median age) infants (Table 1), and explored the underlying drivers for the differences in resistance gene profiles. Finally, we investigated and compared the influence of conventional antibiotic treatment on the infant and adult gut microbiomes, as assessed by changes in microbial composition, antibiotic resistance, and mobile genetic elements, including plasmids.

## Results

### The distribution of ARG profiles in the adult gut is bimodal and reflects the role of *E. coli* as an ARG reservoir

First, we characterized ARGs in the gut microbiome of 217 young adults, aged 18 years, who were members of the COPSAC2000 cohort. A total of 293 ARGs were detected, which conferred resistance to 33 drug classes. In this assemblage, genes associated with resistance to tetracycline and fluoroquinolone were the most abundant (Fig. 1a), followed by those targeting penam, cephalosporin, macrolide, and rifamycin. The main mechanism of resistance encoded by ARGs was antibiotic efflux pumps (Fig. S1). Almost half of all ARGs (42.7%) encoded resistance to at least two different drug classes, and are referred to hereafter as multiple-drug resistance genes (MDR ARGs) (Fig. S1). The most common type of MDR ARG conferred resistance to fluoroquinolone and tetracycline. The majority of ARGs (53% in abundance) in the adult gut originated from Proteobacteria (Fig. S1), specifically from *E. coli* (≈ 40%). The next-largest contribution came from Bacteroidetes, with 31%. Within Proteobacteria, ARG richness was high in several taxa, such as *Escherichia* species, *Pseudomonas aeruginosa*, *Citrobacter braakii*, *Klebsiella pneumoniae*, and *Enterobacter hormaechei* (Fig. 1b). The detailed distribution of ARGs in different bacteria species is shown in Supplementary Data 1. Different bacterial phyla exhibited

## Table 1 | The cohort information in the study

| Main Characteristics | Levels | Statistics |
|---|---|---|
| Cohort | Adults (*N* = 217) | |
| Age, median (range)—yr | | 18 (17–21) |
| Sex—no. (%) | Male \| Female | 101 (46.5%) \| 116 (53.5%) |
| BMI—kg/m$^2$ (mean ± sd) | | 22.4 ± 4.2 |
| Living area | Rural \| Urban | 113 (56.2%) \| 88 (43.8%) |
| Antibiotics in the year—no. (%) | Yes \| No | 51 (24%) \| 166 (76%) |
| Pet—no. (%) | Cat \| Dog \| other | 56 (25.8%) \| 102 (47%) \| 31 (14.3%) |
| Siblings—no. (%) | Yes \| No | 163 (81.9%) \| 36 (18.1%) |
| Smoke—no. (%) | Yes \| No | 93 (43.9%) \| 124 (57.1%) |
| Alcohol Drink—no. (%) | Yes \| No | 202 (93%) \| 15 (7%) |
| Family Income Type—no. (%) | Low \| Medium1 \| Medium2 \| High1 \| High2 | 60 (28.7%) \| 80 (38.3%) \| 52 (24.9%) \| 10 (4.8%) \| 7 (3.3%) |
| Parental Education Level—no. (%) | Low \| Medium \| High | 112 (53.6%) \| 60 (28.7%) \| 37 (17.7%) |
| Cohort | Infants (*N* = 662) | |
| Age, median (range)—yr | | 1 (11month - 2) |
| Sex—no. (%) | Male \| Female | 341 (51.5%) \| 321 (48.5%) |
| Living area | Rural \| Urban | 292 (46.5%) \| 336 (53.5%) |
| Antibiotics in the year—no. (%) | Yes \| No | 311 (47%) \| 351 (53%) |
| Antibiotics during pregnancy—no. (%) | Yes \| No | 271 (40.9%) \| 391 (59.1%) |
| Birth Season—no. (%) | Spring \| Summer \| Autumn \| Winter | 177 (26.7%) \| 139 (21%) \| 141 (21.3%) \| 205 (31%) |
| Pet—no. (%) | Cat \| Dog | 133 (20.2%) \| 123 (18.8%) |
| Gestational age, median (range)—week | | 40 (29 - 42) |
| Fish oil—no. (%) | Yes \| No | 327 (49.5%) \| 334 (50.5%) |
| Siblings—no. (%) | Yes \| No | 382 (72.1%) \| 148 (27.9%) |
| Delivery mode—no. (%) | Vaginal \| Caesarian | 519 (78.4%) \| 143 (21.6%) |
| Breastfeeding—no. (%) | Mixed food \| Only Solid food | 98 (14.8%) \| 562 (85.2%) |
| Housing—no. (%) | House \| Apartment | 229 (42.4%) \| 311 (57.6%) |
| Mother BMI—kg/m$^2$ (mean ± sd) | | 23.6 ± 4.3 |
| Family Income Type—no. (%) | Low \| Medium \| High | 63 (9.5%) \| 352 (53.3%) \| 246 (37.2%) |
| Parental Education Level—no. (%) | Low \| Medium \| High | 53 (8%) \| 423 (63.9%) \| 186 (28.1%) |

Infant family income classification: Low (< €5w), Medium (€5w ~ €11w), High (> €11w); Adult family income classification: Low (< DKK 40w), Medium1 (< DKK 60w), Medium2 (< DKK 80w), High1 (< DKK 100w), High2 (< DKK 200w). Participants with missing information in the "Characteristics" category were excluded from the statistical calculations and relevant analyses.

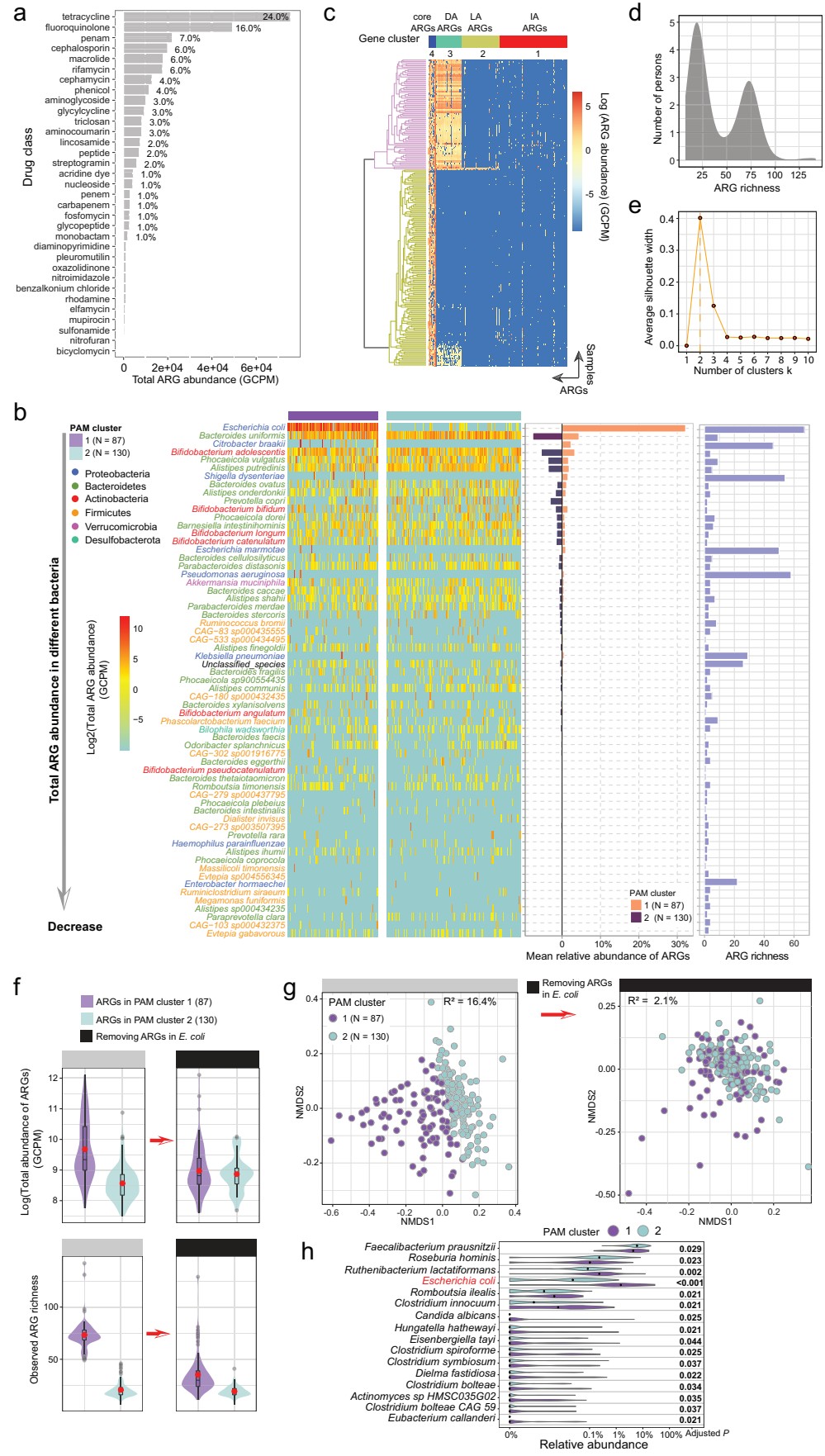

**Fig. 1 | ARG characteristics of different bacteria in the adult gut and bimodal distribution of ARGs in the adult gut, driven by *E. coli*. a** The total abundance of ARGs resistant to 33 drug classes. **b** Total ARG abundance in the bacterial species (left), mean relative abundance of ARGs in bacterial species (middle), and ARG richness in bacterial species (right). For ease of viewing, only the 63 species with the highest ARG abundance are listed. **c** Heatmap depicting the abundance of 293 ARGs across the samples. Samples and ARGs were individually clustered with complete linkage hierarchical clustering and PAM clustering based on Euclidean distance. Cluster 4 (core ARGs, $N = 15$) contained highly abundant and prevalent ARGs across all samples. Cluster 3 (differentially abundant (DA) ARGs, $N = 55$) contained ARGs with significant abundance differences among samples. Cluster 2 (low-abundance (LA) ARGs, $N = 80$) contained ARGs present at a low abundance. Cluster 1 (intermediate-abundance (IA) ARGs, $N = 143$) contained ARGs with intermediate abundance, falling between those in cluster 4 and cluster 2. **d** Density plot of ARG richness in the cohort. **e** Average silhouette width of PAM clustering for k = 1 to 10 clusters. **f, g** Log-transformed total ARG abundance and observed ARG richness (α-diversity) (**f**), and NMDS ordination plot of Bray−Curtis dissimilarity matrix of ARG abundances (**g**) before (left) and after (right) the removal of *E. coli* ARGs from the two ARG PAM clusters ($N_{cluster 1} = 87$, $N_{cluster 2} = 130$). The box plot displays 25th, 50th (median), and 75th percentiles, with whiskers extending 1.5 * IQR. The percent of explained variance ($R^2$) generated with the PERMANOVA test is shown in the figure. **h** Relative abundances of 16 species (of 542 in total) that differed in abundance between the two clusters. Relative abundance on the *x*-axis is shown on a logarithmic scale; black dots indicate median value; *P*-values were generated by the Wilcoxon rank-sum test, and FDR adjustments are represented as adjusted *P*-values. All *P*-values were derived from two-sided tests.

distinct patterns both in terms of the number and type of ARGs present (Fig. S1 and Supplementary Data 2). For example, Proteobacteria contained the highest number of unique ARGs (163), and these were mainly β-lactam resistance genes.

Based on their abundance patterns, ARGs were divided into four non-overlapping groups (Fig. 1c, Supplementary Data 3). Notably, the distribution of ARG richness among samples was bimodal, with one peak with low richness and another peak with high richness (Fig. 1d). Likewise, clustering based on ARG abundance revealed two distinct groups of samples (Fig. 1c): cluster 1 high ARG richness ($n = 87$) and cluster 2 low ARG richness ($n = 130$), which was supported by a "partitioning around medoids" (PAM) clustering analysis (Fig. 1e−g). Compared to cluster 2, ARGs in cluster 1 were not only more abundant but also more diverse (Fig. 1f).

To investigate the factors underlying the bimodal ARG distribution, we compared the bacterial composition of the two clusters. We first determined that there were no differences in sequencing coverage between the samples in the two clusters (Wilcoxon test; $P = 0.21$, Fig. S2), ruling out the influence of sequencing depth. We then characterized bacterial composition using MetaPhlAn[21]. A significant correlation was detected between the composition of bacterial communities and that of ARGs through Procrustes analysis (permutational test; $r = 0.77$, $P = 0.001$, Fig. 2b). Furthermore, the two clusters differed significantly in their bacterial composition (PERMANOVA; $P = 0.001$). To identify which bacteria were critical to this difference, we analyzed differentially abundant bacteria between the two clusters and ranked them according to their importance in shaping the clustering pattern. Among the 542 bacterial species detected, 16 species were differentially abundant between the two clusters (Fig. 1h), and the most important of these was clearly *E. coli*. Indeed, the mean relative abundance of *E. coli* in cluster 1 was 66 times higher than that in cluster 2 (mean relative abundance; cluster 1 vs. cluster 2, 4.55% vs. 0.069%). In addition, random forest analysis demonstrated that *E. coli* content was a determining factor in grouping ARGs, and that it was far more important than any other taxon (Fig. S1).

To investigate this further, we assessed the bacterial origin of ARGs using metagenome-assembled genomes (MAGs). In total, we detected *E. coli* MAGs with ARGs in 112 samples, 86 of which were from cluster 1 and 26 from cluster 2. When we removed these *E. coli*−associated ARGs from all samples, we observed an eight-fold reduction in the proportion of variance explained by the two ARG clusters, from 16.4% to 2.1% (Fig. 1g). Without *E. coli*, ARG abundance and diversity in cluster 1 were significantly lower, to the point that values in the two clusters became comparable (Fig. 1f). This provided clear evidence of the abundance of ARGs in *E. coli* and the effect this has on the overall gut microbiome. Although the mean relative abundance of *E. coli* was only around 1.86% in the adult gut, the mean relative abundance of ARGs in this bacterium accounted for about 32% of the total, with the majority in cluster 1 (Fig. 1b). We tested the overall effects of environmental factors (excluding antibiotics) listed in Table 1 on the adult gut microbiome/resistome. The results showed that these environmental factors did not have a significant effect on the adult microbiome/resistome. However, gender had a significant effect on microbial composition (PERMANOVA; $R^2 = 1.2\%$, adjusted $P = 0.002$).

## ARGs are more abundant in the infant gut than in adults, with *E. coli* as the largest single contributor

We performed a comprehensive comparison of the ARGs described above and those identified, using the same workflow, in a cohort of 662 1-year-old Danish infants.

Overall, ARG profiles were significantly different between adults and infants (β-diversity (Bray−Curtis), PERMANOVA; $R^2 = 8.5\%$, $P = 0.001$, Fig. 2a). Procrustes analysis revealed a significant correlation between the composition of bacterial communities and that of ARGs in both the adult and infant gut (permutational test; r_adults = 0.77, r_infants = 0.78, both $P = 0.001$, Fig. 2b), suggesting that ARG distribution was strongly tied to overall bacterial composition regardless of host age. β-diversity analysis also highlighted a significant difference in gut microbial composition between adults and infants (β-diversity (Bray−Curtis), PERMANOVA; $R^2 = 10\%$, $P = 0.001$, Fig. 2c). Furthermore, of the 896 bacterial species detected, 482 (54%) were differentially abundant between the two cohorts (Wilcoxon test; FDR adjusted $P < 0.05$), indicating that the differences between adults and infants were influenced by the overall bacterial composition. However, considering that *E. coli* contains a large proportion of ARGs in both adults and infants[14] and that the relative abundance of *E. coli* differed between adults and infants (mean relative abundance, infants vs. adults, 5.4% vs. 1.86%, Fig. S3), we wanted to determine whether these age-related differences remained even in the absence of *E. coli*. We thus removed all *E. coli*−associated ARGs from the two groups and re-evaluated the overall differences in ARG composition (Fig. S3). We found that the percentage of variance in ARG profiles that was explained by the two age groups did not decrease in the absence of *E. coli*, indicating that this species is not the only factor shaping age-related differences (Fig. S3). In brief, the dissimilarity in ARG profiles between the two cohorts arises from the concerted influence of the bacterial community, rather than being driven by a solitary bacterium playing a decisive role.

ARGs were more abundant in the infant gut than in the adult gut, as reflected in both the number of ARGs per million genes and the relative abundance of ARGs (Wilcoxon test; $P < 0.001$, Fig. 2d, e). When we removed *E. coli*−associated ARGs from the analysis, the difference between adults and infants in the mean number of ARGs per million genes and the mean relative abundance of ARGs decreased by 53% and 51%, respectively (Fig. 2d, e). These findings imply a significant role of *E. coli* in shaping the divergence of gut ARG load between adults and infants.

Plasmids are important mobile genetic elements that can transfer ARGs between cells. We, therefore, specifically investigated mobile ARGs carried on plasmids in the adult and infant gut. As in the overall analysis, the abundances of plasmids and mobile ARGs were higher in the infant gut than in the adult gut (Wilcoxon test;

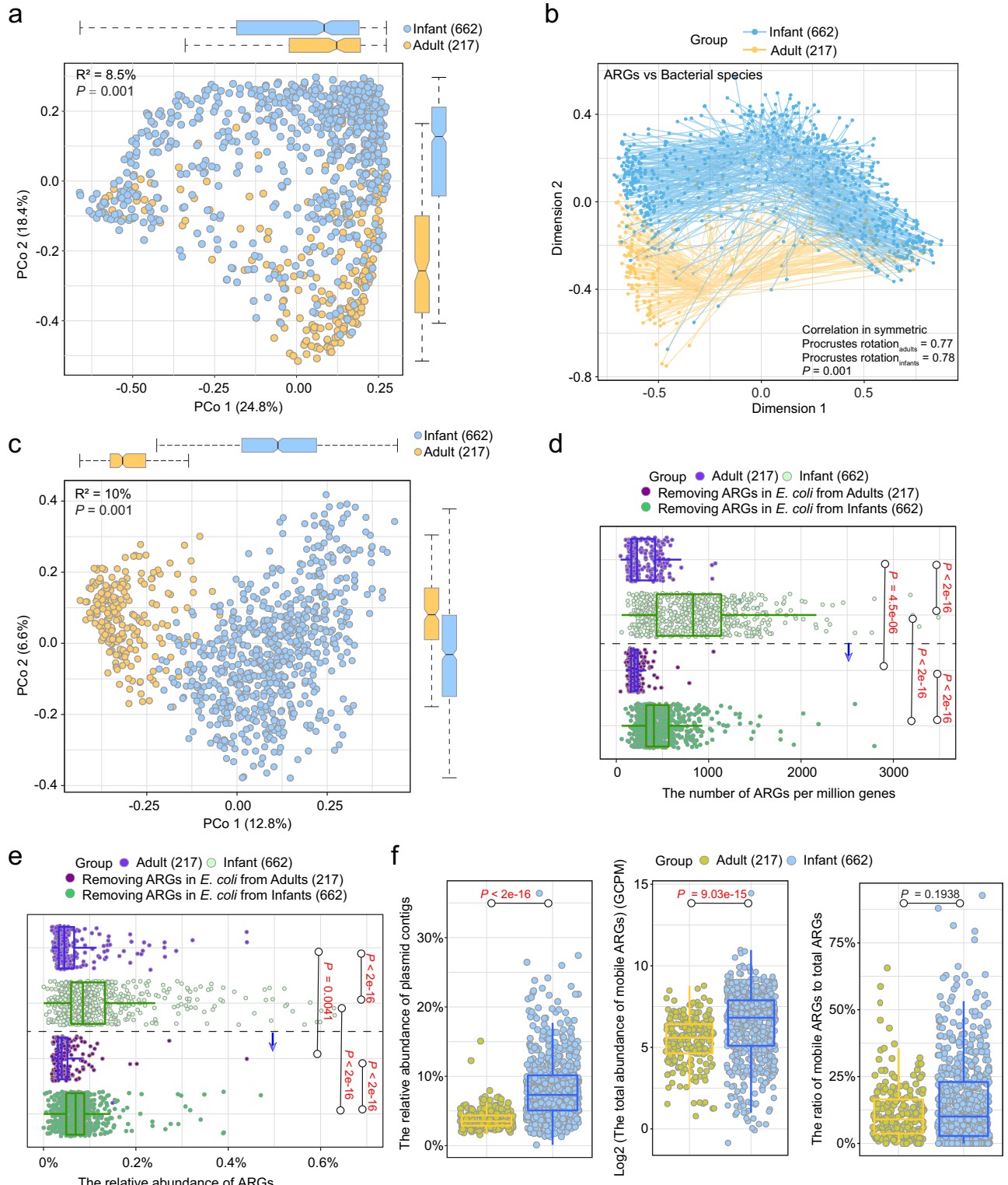

$P < 0.001$, Fig. 2f). However, the proportion of mobile ARGs on plasmids relative to total ARGs did not differ between cohorts (Wilcoxon test; $P = 0.19$, Fig. 2f).

To gain more insight into the ARGs carried by *Escherichia* species in the two cohorts, we plotted phylogenetic trees of *Escherichia* MAGs and clustered MAGs based on their ARG profiles; for the sake of comparison, we also carried out the same procedure for *Bifidobacterium* MAGs. From a phylogenetic perspective, *Escherichia* MAGs

differed between the two cohorts (PERMANOVA; $P = 0.02$, Fig. S4). In addition, *Escherichia* MAGs belonging to four main species correlated with ARG profiles (PERMANOVA; $P = 0.01$, Fig. S4). However, we did not find an ARG profile in *Escherichia* that was exclusive to the adult or infant gut. Instead, in *Bifidobacterium* we found one ARG profile almost exclusively in infants that was also predominantly distributed in one specific MAG cluster (Fig. S5). In addition, many *Bifidobacterium* MAGs did not carry ARGs.

**Fig. 2 | ARG profiles differed significantly between the infant and adult gut, with infants containing a higher abundance of ARGs. a** Comparison of ARG profiles in the adult and infant gut based on Bray–Curtis dissimilarity matrices of ARG abundance ordinated by PCoA plot (values in brackets represent the percentage of variance explained by the principal coordinates). *P*-value and $R^2$ were generated with a PERMANOVA test. Box plots along each axis show the value of each point at the respective coordinates. **b** Procrustes analysis of the association between the composition of ARGs and that of bacterial communities as characterized by MetaPhlAn in the gut of adults and infants. *P*-value was generated with a Permutational test. **c** Comparison of bacterial composition profiles in the adult and infant gut based on Bray–Curtis dissimilarity matrices of bacterial community composition ordinated by PCoA plot. *P*-value and $R^2$ were generated with a PERMANOVA test. Box plots along each axis show the value of each point at the respective coordinates. **d**, **e** Boxplot with jitter points showing the number of ARGs per million genes (**d**) and the relative abundance of ARGs out of all predicted genes by Prodigal (**e**) before and after removing *E. coli* ARGs in the adult and infant gut. *P*-value obtained from the Wilcoxon test and red indicates *P* < 0.05 (significant difference). **f** Boxplot with jitter points showing the relative abundance of plasmid contigs out of all contigs, the log-transformed total abundance of ARGs in plasmids, and the ratio of mobile ARGs to total ARGs in the adult and infant gut. ARGs carried on plasmids are defined as mobile ARGs. *P*-value obtained from the Wilcoxon test and red indicates *P* < 0.05 (significant difference). The box plots (**a**, **c**–**f**) display 25th, 50th (median), and 75th percentiles, with whiskers extending 1.5 * IQR. All *P*-values were derived from two-sided tests.

## Infants and adults share dominant ARGs and bacterial species carrying them in the gut microbiome

Although the overall ARG profiles differed between the infant and adult gut, we wanted to investigate if certain aspects of these assemblages might be shared across age groups. To evaluate this, we explored commonalities between the infant and adult gut in terms of six aspects. First, we compared the alpha diversity (observed richness) of these groups, and found that the number of ARG-carrying bacterial species and the number of mobile ARGs on plasmids were significantly higher in the adult gut than in the infant gut (Wilcoxon test; *P* < 0.001, Fig. S6). We also compared the number and type of ARGs or MGEs in those overlapping ARG-carrying or MGE-carrying bacterial species (Supplementary Data 4), and found the number of MGEs carried by the overlapping MGE-carrying bacterial species was significantly higher in the adult gut than in the infant gut (Wilcoxon test; *P* = 0.012, Fig. S7).

When we identified the ARGs and ARG-carrying bacteria that were overlapping by both infants and adults, we found that they included some of the most abundant representatives in both cohorts. Specifically, infants and adults overlapped 106 ARG-carrying bacterial species, which contributed 68% and 53% of the total ARGs in each group (relative abundance), respectively, while unique species contributed only about 6% of ARGs (relative abundance) (Fig. 3a). Likewise, 191 ARGs were overlapping between the two cohorts, accounting for over 98% of the total ARG abundance in each (Fig. 3b). For the other ARG-related aspects investigated, the results were similar. ARGs and drug-resistance classes that were unique to only one cohort tended to be present in lower abundance (Fig. 3c–f). Details on the comparison of overlapping and unique features with respect to these six ARG-related groups are listed in Supplementary Data 5.

Next, we investigated the top ten drug classes to which these ARGs conferred resistance. For most of these drug classes, infants had a significantly higher abundance of associated ARGs than adults did (Wilcoxon test; adjusted *P* < 0.05, Fig. 3g, h). In both cohorts, tetracycline and fluoroquinolone ARGs were the most abundant. Tetracycline and aminoglycoside were the drug classes most commonly targeted by mobile ARGs in the infant gut, while mobile ARGs in the adult gut more often targeted tetracycline and macrolide.

## Compared to infants, antibiotic treatment in adults had a longer-lasting effect on microbial composition, ARG and MGE profiles, and plasmid abundance

It is well known that antibiotic therapy changes the gut microbiome[17,22], but the extent to which this effect may differ according to age has not yet been characterized. Here, we compared the association between antibiotic treatment and alterations in the gut microbiome in adults and infants. In particular, we examined differences in microbial composition, ARGs, and mobile genetic elements (MGEs), which here included the genetic elements related to mobility, such as integrases, transposases, and insertion sequences. In the adult cohort, the effects of antibiotic treatment persisted up to about 1 year. Instead, for infants, the effects of antibiotic treatment were detectable for about 1 month. Specifically, ARG profiles and microbial community composition were significantly different in the gut of adults who had taken antibiotics within 6 months or between 6 months and 1 year before sampling compared to those who had not (β-diversity, PERMANOVA; <6m, *P* = 0.023, 0.0023, respectively; 6m–1y, *P* = 0.005, 0.03, respectively; Fig. 4a). Instead, MGE profiles differed only in the group that had taken antibiotics within 6 months of sampling (<6m, *P* = 0.03, Fig. 4a). No effects were detectable for any of these three indicators when the antibiotic use had occurred more than 1 year prior to sampling (*P* > 0.05, Fig. 4a). In the infant cohort, ARG and MGE profiles were different in individuals who had received antibiotic treatment within 15 days of sampling or between 15 days and 1 month before sampling compared to those who had not (<15d, *P* < 0.001; 15d–1m, *P* = 0.035, 0.0095, respectively, Fig. 4b). Infants who had taken antibiotics more recently also demonstrated alterations in microbial community composition (<15d, *P* < 0.001, Fig. 4b). None of these effects were apparent if the antibiotic use had occurred more than 1 month before sampling (*P* > 0.05, Fig. 4b).

The duration of the effect of antibiotics in adults and infants was also reflected in plasmid abundance. Plasmids can horizontally transfer resistance and virulence genes between bacterial cells. In the adult gut, the effect of antibiotics on plasmids lasted up to about 1 year: the total abundance of plasmids was higher in the gut of adults who had taken antibiotics within 6 months of sampling or between 6 months and 1 year before sampling than those in the corresponding control groups (Wilcoxon test; *P* < 0.001, Fig. 4c). In contrast, there were no differences in plasmid abundance between adults who had taken antibiotics more than 1 year before sampling and those who had not (Wilcoxon test; *P* > 0.05, Fig. 4c). Similarly, plasmid abundance in the gut of infants who had taken antibiotics more than 1 month before sampling did not differ from those who had not (Wilcoxon test; *P* > 0.05, Fig. 4d). However, plasmids were more abundant in the gut of infants who had received antibiotics between 15 days and 1 month before sampling or within 15 days of sampling than in individuals in the corresponding control groups (Wilcoxon test; *P* = 0.03 (0–15d), *P* = 0.01 (15d–1m), Fig. 4d).

## Antibiotic treatment enhances ARG and MGE abundance and reduces bacterial richness

In addition to the overall alterations, we also observed differences in total ARG and MGE abundance, and bacterial richness as a result of antibiotic treatment. Specifically, ARGs were significantly more abundant in the gut of adults who had taken antibiotics within 1 year of sampling compared with those who had not (Wilcoxon test; *P* < 0.001, Fig. 5a), and the bacterial richness was lower (Wilcoxon test; *P* = 0.022, Fig. 5a). With respect to MGEs, total abundance was higher in adults who had taken antibiotics within 6 months of sampling than in those who had not (Wilcoxon test; *P* = 0.036, Fig. 5a). For infants, the same phenomenon was observed: compared to the corresponding control groups, total ARG abundance was higher in the gut of infants who had taken antibiotics within 1 month of sampling, and gut bacterial diversity was lower in infants who had taken antibiotics within 15 days of sampling (Wilcoxon test; *P* < 0.001, *P* = 0.0048, respectively, Fig. 5b).

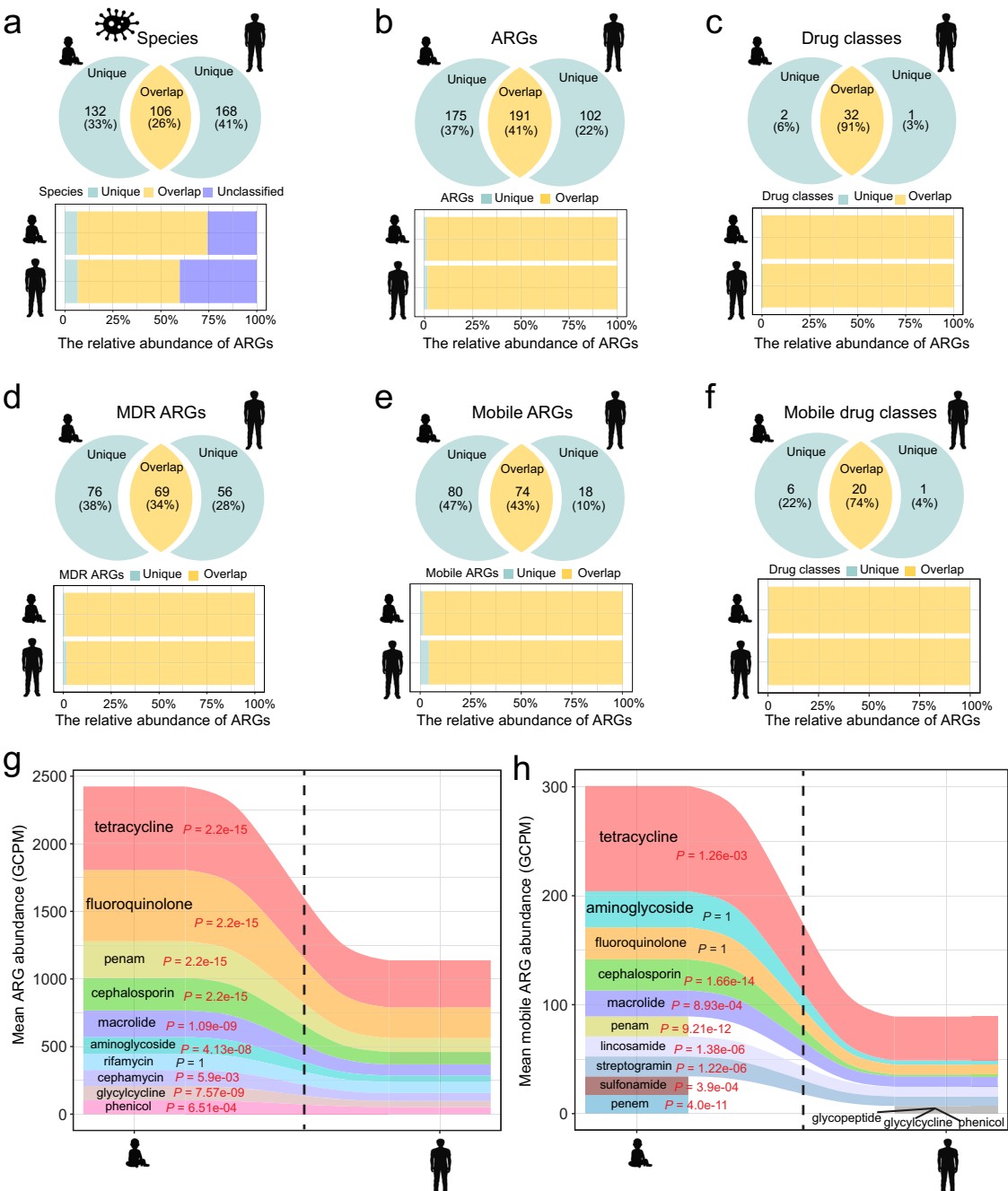

**Fig. 3 | ARGs overlapping by the adult and infant gut accounted for the vast majority of ARG abundance in each cohort.** Analyses of the unique and overlapping (**a**) ARG-carrying bacterial species, (**b**) ARGs, (**c**) drug classes targeted by ARGs, (**d**) MDR ARGs, (**e**) mobile ARGs, (**f**) and drug classes targeted by mobile ARGs in both gut, with respect to the number of individual species/genes/drug classes (top panel) and their relative abundance in the total population of ARGs (bottom panel). "Unique" represents species/ARGs/drug classes that were only present in adult or infant gut regardless of prevalence and abundance. "Unclassified" represents ARGs with unknown bacterial origin, as these ARG-carrying contigs were not detected within the bins. Mean abundance of the 10 most commonly targeted drug classes by ARGs (**g**) and by mobile ARGs (**h**) in the adult and infant gut. *P*-value from the Wilcoxon test with Bonferroni adjustment and red indicates *P* < 0.05 (significant difference). Seven of the 10 mobile drug classes were shared between cohorts. All *P*-values were derived from two-sided tests.

We then explored the effects of antibiotics on the abundance of different types of ARGs: specifically, the four groups of ARGs in the adult gut, clustered using the PAM algorithm (core, DA, IA, and LA; Fig. 1c, Supplementary Data 3) and three clusters in the infant gut, obtained using the same methodology (Fig. S8). We found that antibiotic treatment enhanced the total abundance of low-abundance ARGs in adults and intermediate-abundance ARGs in infants (Wilcoxon test; adjusted *P* = 0.044, *P* < 0.001, respectively,

Fig. 5c, d). Interestingly, the total abundance of core ARGs—resistance genes that are highly abundant and prevalent overall—also increased in the gut of both adults and infants after antibiotic treatment (Wilcoxon test; adjusted *P* < 0.001, 0.015, respectively, Fig. 5c, d). The mean abundance of most individual core ARGs was higher in individuals who had taken antibiotics than in those who had not, although this was not statistically significant (Wilcoxon test; adjusted *P* > 0.05, Fig. S9).

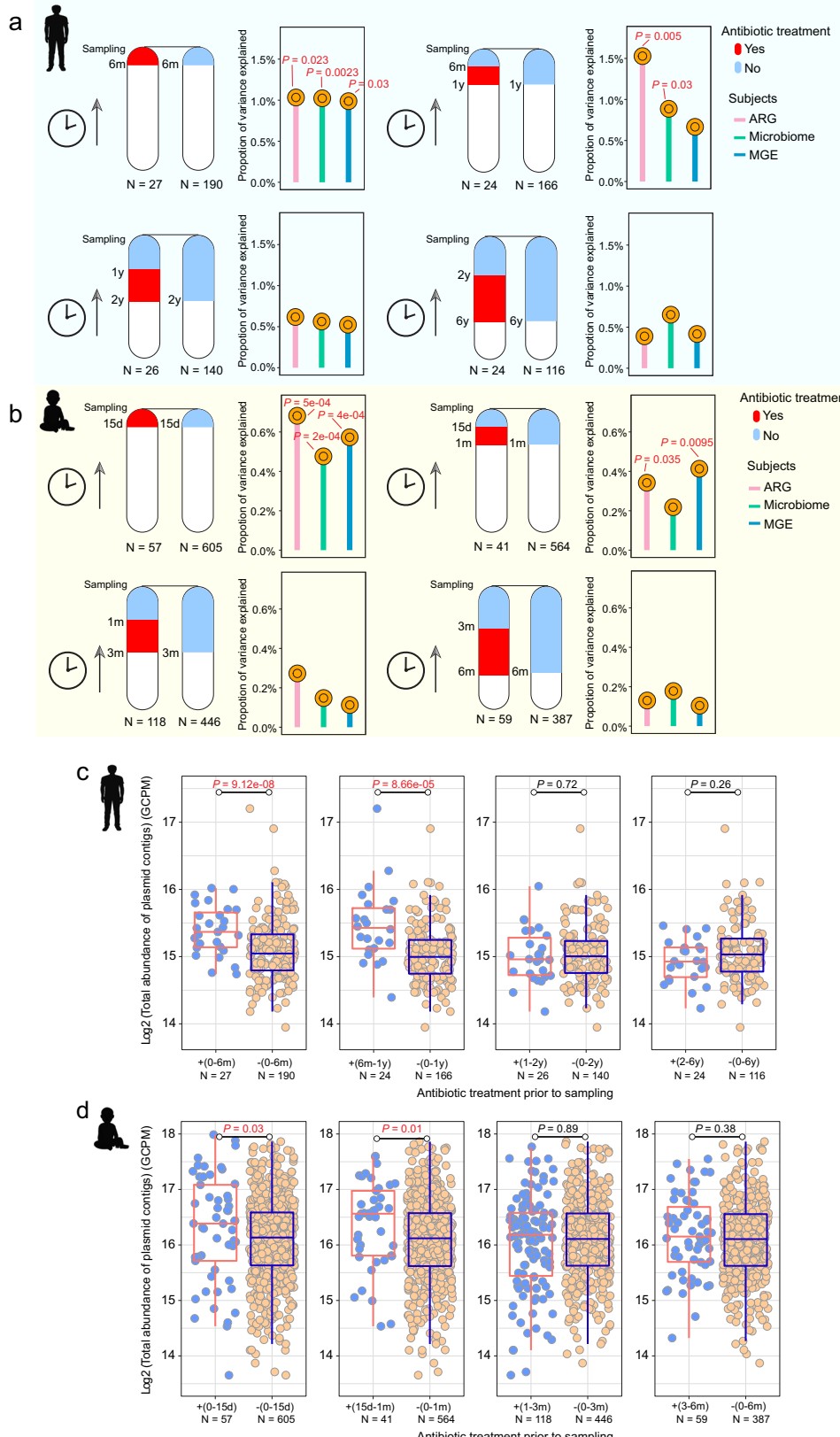

Fifteen core ARGs, mostly associated with tetracycline and MLS resistance (Fig. S9), were detected in the adult gut and were found in between 54% and 100% of samples (mean 76.2%). For several of these ARGs—specifically, *ErmB/H/G*, *tet(40)/O/Q/W*, and *vanI*—more than 20% of these genes were retrieved from plasmids. Two core ARGs (*adeF* and *tetQ*) were detected in 97.7% and 85.8% of the infant gut samples, respectively, and 36% of the latter appeared on plasmids (Fig. S9).

**Fig. 4 | Antibiotic treatment had longer-lasting effects on the adult gut microbiome than on the infant gut microbiome, as reflected in microbial composition, ARG and MGE profiles, and plasmid abundance.** Duration of the effect of antibiotic administration on the β-diversity (Bray–Curtis distance) of microbiome, ARG and MGE compositions in the adult gut (**a**) and in the infant gut (**b**). Adult subjects were divided into four groups depending on when they had taken antibiotics: within 6 months of sampling, 6 to 12 months prior, 1 to 2 years prior, or 2 to 6 years prior to sampling; the corresponding control groups had not received antibiotics in those periods. Infant subjects were divided into four groups depending on when they had taken antibiotics: within 15 days of sampling, 15 to

30 days prior, 1 to 3 months prior, and 3 to 6 months prior; the corresponding control groups had not received antibiotics in those periods. *P*-value obtained from the PERMANOVA test and red indicates *P* < 0.05 (significant difference). Duration of the effect of antibiotic administration on total plasmid abundance in the adult gut (**c**) and in the infant gut (**d**). The four studied periods are the same as in **a** or in **b**. "+" represents antibiotics administered in a given period, and "−" represents antibiotics not administered in a given period. *P*-value from the Wilcoxon test and red indicates *P* < 0.05 (significant difference). The box plots (**c**, **d**) display 25th, 50th (median), and 75th percentiles, with whiskers extending 1.5 * IQR. All *P*-values were derived from two-sided tests.

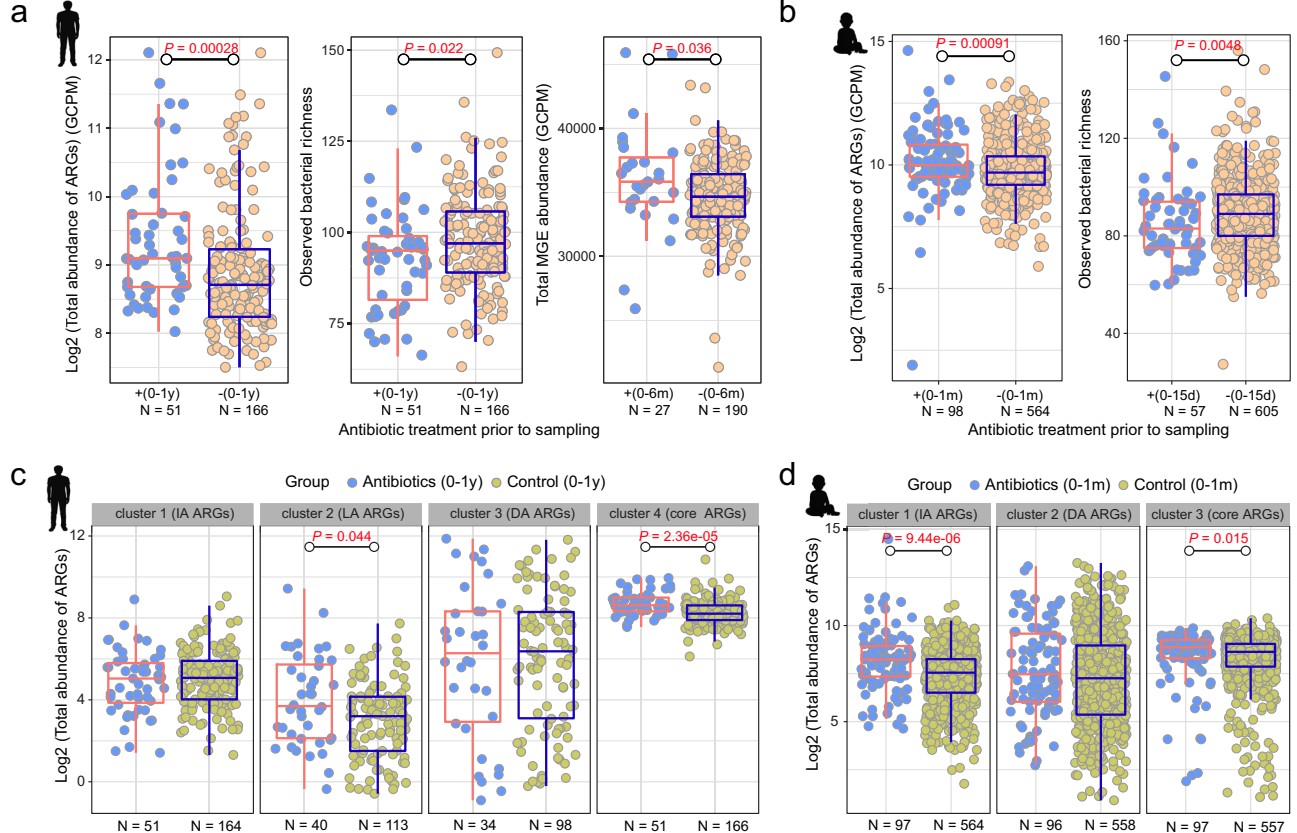

**Fig. 5 | Antibiotic treatment resulted in an elevated abundance of ARGs and MGEs, and a decrease in observed bacterial richness. a** Changes in ARG abundance and bacterial diversity in the gut of adults who had taken antibiotics within 1 year of sampling and changes in MGE abundance in the gut of adults who had taken antibiotics within 6 months of sampling. Individuals who had not taken antibiotics during those periods were used as controls. *P*-value obtained from the Wilcoxon test and red indicates *P* < 0.05 (significant difference). **b** Changes in ARG abundance in the gut of infants who had taken antibiotics within 1 month of sampling and changes in bacterial diversity in the gut of infants who had taken antibiotics within 15 days of sampling. Individuals who had not taken antibiotics during those periods were used as controls. *P*-value obtained from the Wilcoxon test and red indicates *P* < 0.05 (significant difference). Changes in the abundance of ARG clusters in the gut of adults (**c**) who had taken antibiotics within 1 year of sampling and in the gut

of infants (**d**) who had taken antibiotics within 1 month of sampling. Individuals who had not taken antibiotics during those periods were used as controls. For adults, the definitions of these four groups and the methodological basis for clustering are described in the legend of Fig. 1c. For infants, ARGs were clustered into three categories by PAM clustering based on Euclidean distance (Fig. S8); Cluster 3 (core ARGs, *N* = 2) contains highly abundant and prevalent ARGs. Cluster 2 (differentially abundant (DA) ARGs, *N* = 55) contains ARGs with significant differences in abundance between samples. Cluster 1 (intermediate-abundance (IA) ARGs, *N* = 309) contains ARGs whose abundance in the samples falls between the ARGs in cluster 3 and those in cluster 2. *P*-value obtained from the Wilcoxon test with FDR adjustment and red indicates *P* < 0.05 (significant difference). The box plots (**a**–**d**) display 25th, 50th (median), and 75th percentiles, with whiskers extending 1.5 * IQR. All *P*-values were derived from two-sided tests.

## The influence of different antibiotics on the gut microbiome of adults and infants

In the group of adults who had received antibiotic treatment in the year before sampling, we examined whether the type of antibiotic taken had a detectable influence on characteristics of the gut microbiome compared to control groups. Except for β-lactam plus sulfonamide, each type of antibiotic was associated with an increase in the mean abundance of ARGs, with tetracycline and β-lactam plus

macrolide having a statistically significant effect (Wilcoxon test; adjusted *P* = 0.036, 0.029, respectively, Fig. 6a). Each antibiotic type was also associated with an increase in mean plasmid abundance, with β-lactam, tetracycline, and β-lactam plus macrolide having statistically significant effects (Wilcoxon test; adjusted *P* = 0.049, 0.038, 0.00051, respectively, Fig. 6b). Four of the five antibiotic types were also associated with a reduction in mean bacterial richness (exception was β-lactam plus sulfonamide, Fig. S10), and all five antibiotics were

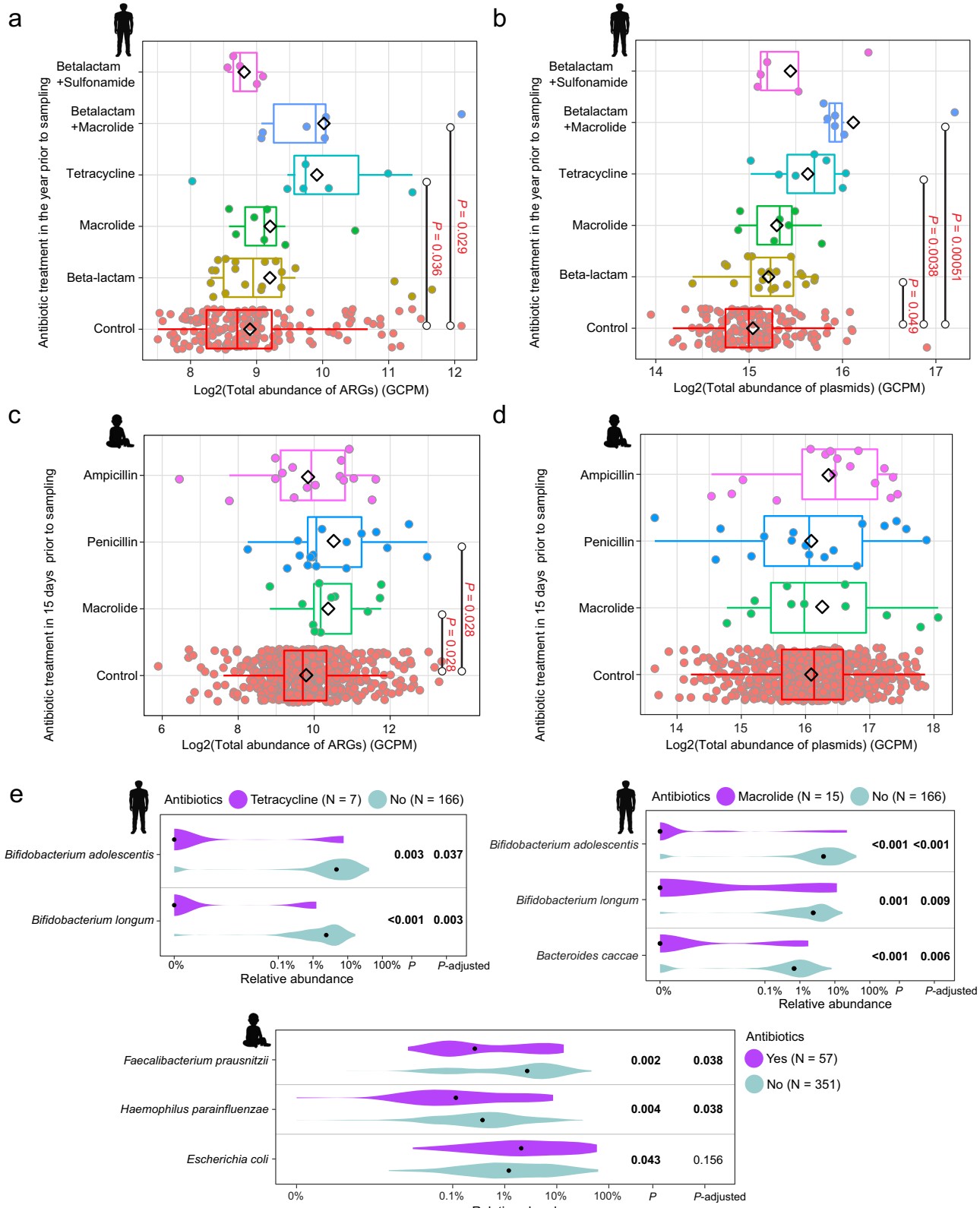

associated with increases in mean MGE abundance (Fig. S10). Finally, treatment with tetracycline or macrolide resulted in a significant reduction in the relative abundance of *Bifidobacterium adolescentis* and *Bifidobacterium longum*, two of the 20 most abundant species (Wilcoxon test; adjusted *P* < 0.05, Fig. 6e).

In the infant cohort, we evaluated whether treatment with one of three major antibiotics—macrolide, penicillin, and ampicillin—in the 15

days before sampling had distinguishable effects on the infant gut microbiome. All antibiotics were associated with an increase in mean ARG abundance, with macrolide and penicillin having a statistically significant relationship (Wilcoxon test; adjusted *P* = 0.028, 0.028, respectively, Fig. 6c). Furthermore, all antibiotics were associated with non-significant increases in mean plasmid abundance (Wilcoxon test; adjusted *P* > 0.05, Fig. 6d) and reductions in mean bacterial

**Fig. 6 | The effects of different antibiotics on ARG and plasmid abundance, and on the relative abundance of bacterial species.** Changes in ARG abundance (**a**) and plasmid abundance (**b**) in the gut of adults who had taken one of five major antibiotics or antibiotic combinations in the year before sampling (N = 5, 6, 7, 7, 20, respectively). Individuals who had not taken antibiotics in that period were used as controls (N = 166). P-value obtained from the Wilcoxon test with FDR adjustment and red indicates P < 0.05 (significant difference). The black diamond indicates the mean value. Changes in ARG abundance (**c**) and plasmid abundance (**d**) in the gut of infants who had taken one of three major antibiotics in the 15 days before sampling (N = 16, 17, 11, respectively). Infants who had not taken antibiotics within 15 days of sampling were used as controls (N = 605). P-value obtained from the Wilcoxon test with FDR adjustment and red indicates P < 0.05 (significant difference). The black diamond indicates the mean value. The box plots (**a**–**d**) display 25th, 50th (median), and 75th percentiles, with whiskers extending 1.5 * IQR. **e** Members of the 20 most abundant bacterial species whose abundance in the gut differed significantly between (top) adults who had taken tetracycline or macrolide in the year before sampling and those who had not received antibiotic treatment, and (bottom) infants who had taken antibiotics (mixed effects) in the 15 days before sampling and those who had not within the first year. Relative abundance on the x-axis is shown on a logarithmic scale; black dots indicate median value; P-values were generated by the Wilcoxon rank-sum test and adjusted using FDR. All P-values were derived from two-sided tests.

richness (Fig. S10). Macrolide and penicillin were linked with increases in mean MGE abundance (Fig. S10). None of the three antibiotics had a statistically significant influence on the abundance of the 20 most abundant bacterial species. When we investigated the mixed effect of antibiotics on the broader bacterial community, we found that antibiotics were associated with a significant decrease in the relative abundance of *Faecalibacterium prausnitzii* and *Haemophilus parainfluenzae* (Wilcoxon test; adjusted P < 0.05, Fig. 6e). Additionally, we observed an increase in the abundance of *E. coli*, although the adjusted P value was not significant.

## Discussion

Metagenomic sequencing offers the possibility to gain deeper insight into the distribution and function of ARGs in gut microbes at the species or strain level. Using this approach, we examined the distribution of ARGs in the gut bacteria of 217 young Danish adults, aged 18 years. By combining this information with similar data from 662 1-year-old Danish infants, we were able to describe age-related patterns in the abundance and distribution of ARGs in the gut, as well as associations between antibiotic use and alterations in the gut microbiome, ARGs, and MGEs, including plasmids, across age groups.

In the adult cohort, we obtained evidence that ARGs follow a bimodal distribution that is driven by the abundance of *E. coli*. A similar bimodal distribution had been found for ARGs in the infant gut[14], which suggests that this phenomenon is independent of age. Numerous genomic/molecular studies and in vitro resistance assays have shown that members of family *Enterobacteriaceae* possess an extremely broad array of antibiotic resistance[23–25], particularly to beta-lactams, which has largely been attributed to gene flow under sustained selective pressure resulting from the increase in antibiotic use in recent decades[26,27]. In both the adult and infant gut, the ARG profiles on *Escherichia* MAGs were quite similar, providing additional evidence for the frequent influx of genes into the *Escherichia* genome. Moreover, many studies have shown that this gene transfer is not unidirectional: the rich pool of resistance elements in *Enterobacteriaceae* genomes also flows to other bacteria[28,29], thereby exacerbating the spread of resistance genes.

Although our study is not longitudinal, it does provide a cross-sectional view of the differences in gut ARGs between early life and adulthood in the Danish population. We discovered that the dominant ARGs, and the bacterial species on which they were found, were the same in both infants and young adults, which could indicate a prolonged selective advantage or a shared community reservoir. Such a selective advantage, i.e., the persistence of certain genes or gene-carrying bacteria throughout childhood, would likely be due to ongoing selection from external factors such as repeated antibiotic therapy[30,31] and/or a competitive advantage over their bacterial neighbors.

Compared to infants, the proportion, number, and abundance of ARGs was lower in the adult gut, and this was associated with decreased levels of clinically relevant bacteria that contain abundant resistance genes, such as *E. coli* and *Shigella flexneri*. *Shigella flexneri* are pathogenic bacteria that can cause bacillary dysentery or shigella dysentery[32]. This mirrors previous findings that infants have a higher load of resistance genes in their gut compared to their mothers[33]. Similar results have even been reported from cattle and pigs, in which the abundances of ARGs and resistance-carrying *Enterobacteriaceae* in the gut are also high early in life and decline with age[34]. Importantly, this early-life peak in *Enterobacteriaceae* does not seem to be driven by any external factors such as antibiotic use; instead, its trajectory in the gut may be related to favorable environmental conditions and host regulation. Facultative anaerobes such as *E. coli* can consume oxygen and produce an anaerobic environment, thus favoring subsequent colonization by and growth of strictly anaerobic bacteria[35]. Previous studies have highlighted various mechanisms by which a host can manage the development of the gut microbiome, such as the immune system response[36], the production of nitrogen-rich mucins, and the creation of a more suitable habitat[37,38]. Abundance of ARG-carrying bacteria may cause a delayed maturation of gut microbiome and increase the risk of asthma later in life[14]. Obviously, such enrichment poses a threat to infant health by reducing the effectiveness of antibiotic therapy for bacterial infections[39]. Our observation that plasmids were abundant in the infant gut also implies a high frequency of HGT[40,41] which can provide an advantage for the dissemination and persistence of ARGs even in the absence of antibiotics[42].

Compared to adults, though, the gut microbiome in infants recovered more quickly from antibiotic therapy. The infant gut microbiome is very dynamic[43] and less diverse than that of adults, which may indicate that the ecological processes at play are simpler and can more easily recover from perturbations. However, this effect is also mediated by the types and doses of antibiotics used[44–46]. In Denmark, the type and dose of common antibiotics vary according to age[47]. Moreover, the length of the recovery period after antibiotic treatment has also been found to depend on the disease targeted. The present study examined the effects of routine antibiotic treatment on common infections. Instead, in neonates with sepsis or extremely preterm infants who were treated with broad-spectrum antibiotics, the overall gut microbiome took a long time to return to normal[18,48]. In examining potential confounders of antibiotic use in infants, we found that infants whose mothers had taken antibiotics during pregnancy were much more likely to take antibiotics during their first year of life (Fisher's test; odds ratio = 1.41, P = 0.03). It is important to note that our analysis examined the mixed effect of all antibiotics taken, where the effects of additional antibiotics may confound the results. Furthermore, although our results indicated that the infant gut microbiome typically returned to baseline levels after about 30 days, we cannot rule out some potential long-term effects that were not addressed in our analysis, such as alterations in specific resistance genes and bacteria[49], immune maturation[50], or metabolic changes[51]. This requires further research using longitudinal samples. In addition, we cannot rule out confounding by indication—that the antibiotic-treated vs. non-treated infants and adults differed due to factors that contributed to the condition their treatment was prescribed for.

The total abundance of core ARGs was significantly elevated in both the infant and adult gut following antibiotic exposure, implying that they are the primary weapons of bacteria against antibiotics and thus possess the potential for widespread dissemination. This was also

supported by the patterns we identified in high ARG prevalence and abundance, as well as plasmid presence. However, different antibiotics had different effects on the abundance of both ARGs and plasmids. Of the five major antibiotics used in adults, tetracycline and beta-lactam plus macrolide had the strongest impact on ARG and plasmid abundance. The effect of the former may be related to the extreme abundance of tetracycline resistance genes in bacteria and plasmids in the adult gut. Although the medical use of tetracycline has declined over the past 20 years and it is no longer recommended to treat pregnant women and children under 8 years of age[52], it remains one of the most widely used classes of antibiotics worldwide[53]. With respect to the latter, there may be a synergistic effect of taking separate courses of beta-lactam and macrolide within a year which simultaneously calls into action resistance genes against both beta-lactam and macrolide as well as plasmids carrying relevant genes in the gut. In infants, the administration of penicillin or macrolide in the 15 days prior to sampling was significantly associated with high ARG abundance. In previous work, we found that the influence of macrolide treatment on macrolide resistance genes in the infant gut could last for approximately 2 months, whereas the effect of penicillin was much shorter[14]. A study on Finnish children (2–7 years, median age 5 years) also confirmed that macrolide treatment had a stronger impact on the gut microbiome than penicillin did[45]. In the adult gut, both tetracycline and macrolide were associated with dramatically reduced levels of the beneficial bacteria *Bifidobacterium adolescentis* and *Bifidobacterium longum*, which are the most prevalent *Bifidobacterium* species in the adult gut[54,55] and are effective degraders of plant-derived fructooligosaccharides[56]. Similarly, antibiotic administration in infants was found to reduce gut levels of *Haemophilus parainfluenzae*, a conditionally pathogenic bacterium that can cause multiple infections[57,58], but simultaneously reduced levels of *Faecalibacterium prausnitzii*, which is widely considered to be beneficial to host health[59,60]. This reflects the double-edged nature of antibiotic treatment, which kills pathogenic bacteria to cure disease but can also kill sensitive beneficial bacteria. Therefore, the type of antibiotic used, and its potential double-edged effects, should be fully considered in the choice of antibiotic treatment.

## Methods

### Human samples

The COPSAC$_{2000}$ cohort is a mother-child cohort assembled for the primary purpose of studying asthma[61]. The 217 fecal samples used for this study were collected as part of the 18-year follow-up visit at the research clinic following detailed instructions. The 662 fecal samples were obtained from 1-year-old infants in the COPSAC$_{2010}$ cohort[62]. Upon arrival at the laboratory, every infant sample was blended with 1 mL of glycerol broth at a concentration of 10% vol/vol and subsequently stored at −80 °C prior to DNA extraction.

### Ethics

The study was designed with the guiding principles of the Declaration of Helsinki in mind and was approved by the Local Ethics Committee of the Danish Capital Region (COPSAC2000: KF 01-289/96, COPSAC2010: (H-B-2008-093)) and the Danish Data Protection Authority (both cohorts: 2015-41-3696). Both parents gave written informed consent for the use of samples and data for this study before enrollment.

### Covariates

During scheduled visits to COPSAC clinics, information was collected from participants on the use of antibiotics (including any treatment prior to sampling), the use and duration of other medications, pet ownership, siblings, living area, income, alcohol consumption, smoking, and experiences with disease. This information was verified against registration records.

### Metagenomic sequencing of fecal samples and data processing

Genomic DNA was extracted from fecal samples (~200–250 mg) using the NucleoSpin® 96 Soil DNA Isolation Kit optimized for epMotion® (Macherey-Nagel, Düren, DE) using the epMotion® robotic platform model (Eppendorf) following the manufacturer's protocol. The feces were weighed within a clean bench in order to control potential contamination from the environment. DNA library preparation and data processing were carried out for adult samples following the same protocol used for infant samples[14]. In brief, the DNA library was prepared for Illumina sequencing with the Kapa HyperPrep kit (KAPA Biosystems, Wilmington, MA, USA). Paired-end (150 bp) sequencing of the samples in the DNA library was performed with the Illumina NovaSeq platform by Novogene (China). Bioinformatics analyses were executed in parallel using GUN parallel v20180722[63]. Adapters were removed using BBDuk of BBTools v38.19 (sourceforge.net/projects/bbmap/). Sickle v1.33[64] was used to trim quality reads with Sanger quality values, with a default quality threshold of 20, and the minimal length threshold of 100bp for the resulting reads after trimming. Human DNA was filtered out using BBMap of BBTools v38.19. In total, 217 gut samples were successfully sequenced, generating between 52.9 and 103 million clean reads per sample (mean ± SD: 58.9 ± 4.5 million reads). The average metagenomic coverage and sequence diversity for each sample were estimated using Nonpareil v3.30 in kmer mode[65]. The mean coverage of adult and infant metagenomic data was 96.42% and 98.23%, respectively (Fig. S11), which represented "almost complete coverage" (≥95% of mean coverage). The species-level composition of microbial communities was described using MetaPhlAn v2.7.5[21]. Sequence assembly was performed with SPAdes v3.12.0 using default metagenomic settings[66]. Bins were created using Variational Autoencoders for Metagenomics Binning (VAMB)[67], a method that uses deep learning to bin microbial genomes. All metagenome-assembled genomes (MAGs) at least 200 kbp in length were submitted for taxonomic assignment with the GTDB-Tk v1.7.0 toolkit, based on the GTDB database (release 202)[68]. Among them, the taxonomy of 84.4% big MAGs in 1250 clusters was assigned, which can cover 70% of contigs in MAGs. Genes were predicted with Prodigal v2.6.3 in META mode[69]. The reads assigned to *E. coli* by MetaPhlAn were subdivided into two main MAGs, one for *E. coli* and the other for *Shigella flexneri* (aka *E. flexneri* in GTDB). For consistency, the analyses in Figs. 1 and 2 involving *E. coli* MAGs were a merger of the two.

### ARG and MGE prediction and gene abundance calculation

Resistance gene identifiers (RGI) were used to annotate ARGs based on the Comprehensive Antibiotic Resistance Database (CARD v3.0.7)[70]. ARGs with the strict and perfect thresholds of the RGIs were kept for further analysis. MGE homologs were characterized by HMM search in HMMER3 v3.1b2[71] in combination with the PFAM[72] and TnpPred[73] databases, with "cut_ga" as a threshold criterion[74,75]. If multiple MGE alignments were detected for one gene, only the one with the lowest E value was kept.

Reference genes were indexed using bowtie2-build of Bowtie2 v2.3.5 before aligning reads[76]. Clean reads were aligned against the predicted genes with Bowtie2 aligner. The number of mapped reads in bam files was calculated with Samtools idxstats of Samtools v1.12[77]. Values of gene coverage per million (GCPM)[14], which normalize sequencing depth and gene length, were used to quantify gene abundance. The sum of the GCPM values for all predicted genes in each sample was one million, making it comparable across samples. The formula for calculating GCPM for each gene is $\frac{(\text{counts}/\text{gene length}) \times 10^6}{\sum_1^n \text{counts}/\text{gene length}}$, where counts are the number of mapped reads, gene length is the length of the gene, and n is the total number of the predicted gene in each sample.

## Plasmid prediction and calculation of contig abundance

Plasmid contigs were identified and characterized with Platon v5.3 using the default settings[78]. Reference contigs were indexed using bowtie2-build before aligning reads. Clean reads were aligned against the contigs with Bowtie2 aligner. The number of mapped reads in bam files was calculated with Samtools idxstats. GCPM values were used to quantify contig abundance as described above. The sum of the GCPM values for all contigs in each sample was one million, and the formula for calculating GCPM for each contig is $\frac{(counts/contig\ length) \times 10^6}{\sum_1^n counts/contig\ length}$, where counts is the number of mapped reads, contig length is the length of the contig, and n is the total number of the contigs in each sample.

## Relative importance of bacterial species as evaluated by Random Forest

The relative importance of bacterial species in shaping ARG clusters was evaluated by Random Forest analysis[79] using the R-package "randomForest" v4.7.1.1[80]. The number of trees (ntree) and the number of variables per split (mtry) in the random forest model were set to 500 and 50, respectively, resulting in a stable classifier and a low error rate of 5.99%. The mean decrease in Gini value associated with a predictor was used to estimate the importance of a bacterial species; a higher value indicates a higher importance for that variable.

## Comparing ARG and bacterial distributions using Procrustes analysis

Procrustes analysis was used to evaluate the association between the distribution of microbial species and the distribution of ARGs in each sample[81]. A Hellinger transformation was first performed on the ARG matrix and the species abundance matrix, respectively. Bray–Curtis dissimilarity values were calculated between all samples in the two matrices using the R function "vegdist" in the "vegan" package, v2.6.2. PCoA ("phyloseq" package v1.38.0) was used to ordinate each dissimilarity matrix. The two ordinated dissimilarity matrics were rotated with the R function "procrustes" in the "vegan" package. The R function "protest" in the "vegan" package was used to calculate the symmetric Procrustes correlation coefficient r, the sum of squared distance, and a P-value with 9999 permutations. The association between the distribution of microbial species and ARGs was visualized with ggplot2.

## Construction of phylogenetic tree of metagenome-assembled genomes (MAGs)

The nucleotide-level similarity between MAGs assigned to *Escherichia* or *Bifidobacterium* was assessed with average nucleotide identity (ANI) values using FastANI v1.33[82]. We then used the neighbor-joining method to construct phylogenetic trees[83]. Based on the presence or absence of ARGs in the contigs, the PAM clustering method was used to group *Escherichia* and *Bifidobacterium* MAGs into four categories each, represented by different colored branches. MAGs assigned to *Escherichia* and *Bifidobacterium* belonged to a total of seven and eight metagenomic species, respectively. The dissimilarity between MAGs was quantified using the cophenetic distance. Permutational multivariate analysis of variance (PERMANOVA) was used to investigate differences in cophenetic distances between MAG clusters based on ARG profiles or between MAGs (R-package "vegan" v2.6.2)[84]. With respect to genus *Escherichia*, MAGs from the four main species—*E. coli*, *E. coli_D*, *E. flexneri*, and *E. dysenteriae*—were included in the statistical analysis.

## α-diversity and β-diversity

All data processing and statistical analyses were carried out using the open-source statistical program R. The observed richness of ARGs and bacterial species was used to assess within-individual diversity (α-diversity), while the Bray–Curtis index served as a measure of between-individual diversity (β-diversity). The ordination of β-diversity matrices was performed with NMDS or PCoA (R-package "phyloseq" v1.38.0)[85]. The Wilcoxon rank-sum test was used to test for differences in α-diversity among groups (R package "stats" v4.1.2). PERMANOVA was used to investigate differences in β-diversity (the percent of variance explained can be obtained from outcomes). Adjustments were made for multiple comparisons using the Benjamini–Hochberg correction.

## Partitioning Clustering for samples or ARGs based on ARG composition

Cluster analyses of samples or ARGs based on ARG composition were performed with Partitioning Around Medoids (PAM) clustering[86] using the R function "pam" in package "cluster" v2.1.3[87]. The average silhouette width, which serves as an estimate of the average distance between clusters, was used to assess the quality of PAM clustering; a larger value means better clustering. Euclidean distance was applied to the PAM clustering analysis. The R function "fviz_nbclust" in package "factoextra" v1.0.7[88] was used to determine and visualize the optimal number of PAM clusters.

## Differential abundance analysis

Wilcoxon rank-sum tests were used to identify the bacterial taxa that were differentially abundant between two groups, with multiple tests corrected by FDR. Likewise, ARG, MGE, and plasmid abundances were compared between two groups using the Wilcoxon rank-sum test with FDR correction. Information on the two groups used for comparison has been noted in the context or the figure legends.

## Linear regression analysis

A linear model (R function "lm") was fitted to investigate the extent to which the abundance of *E. coli* explained the variance in the number of ARGs per million genes and the relative ARG abundance. The normality assumption of residuals was checked using the QQ plot.

## Statistics and reproducibility

No statistical method was used to predetermine sample size. The experiments were not randomized due to inapplicability. Samples with DNA concentration below 1 ng/ul which is a minimum concentration for sequencing library preparation were excluded as their DNA was deemed unreliable and defined as failed. Researchers were not blinded during data acquisition and analysis due to inapplicability. All statistical analyses were conducted in R version 4.1.2. The figures or figure legends specify the number of samples used in each statistical analysis.

## Reporting summary

Further information on research design is available in the Nature Portfolio Reporting Summary linked to this article.

## Data availability

The COPSAC2010 metagenomics datasets are available in the Sequence Read Archive (SRA) under the accession number PRJNA715601. The COPSAC2000 metagenomics data have been deposited in the SRA under the accession number PRJNA916259. The MAGs generated from both cohorts have been deposited at DDBJ/ENA/GenBank under the accession number PRJNA1026956. According to the Danish Data Protection Act and European Regulation 2016/679 of the European Parliament and the Council (GDPR), data involving the personal privacy of project participants cannot be publicly available. Research collaborations are open, and data can be accessed via joint research collaborations by contacting the COPSAC Data Protection Officer, Dr. Ulrik Ralfkiaer, at administration@dbac.dk. All other data that support the results of this study has been uploaded to http://mibi.galaxy.bio.ku.dk/R_script and Source_data/.

## Code availability

The R code for data analysis and source data can be found at http://mibi.galaxy.bio.ku.dk/R_script and Source_data/.

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

## Acknowledgements

We appreciate the commitment and assistance provided by the children and families who participated in the COPSAC cohort study. We also recognize and value the special contributions made by each member of the COPSAC research team. Metagenomics sequencing was supported via Novo Nordisk Foundation Grant no. NNF19OC0057934598, Novo Nordisk Foundation Grant no. NNF17OC0025014, and Research Council of Norway project no. 300489. X.L. was supported by BIOCODEX International Grant 2022. Metagenomics analysis was performed by Computerome.

## Author contributions

X.L., S.J.S., and M.A.R. conceived the project. M.A.R., J.S., and J.T. collected the samples and information about various environmental exposures. X.L., A.B., T.Z, J.R., and G.A.V. performed metagenomics and statistical data analysis. X.L. wrote the paper. M.A.R., J.T., J.S., A.B., J.R., and U.T. helped interpret the data. All authors read, revised, and approved the final manuscript.

## Competing interests

The authors declare no competing interests.
