## [Peer Review File · Nature Communications]

REVIEWER COMMENTS

Reviewer #1 (Remarks to the Author):

SUMMARY

In their manuscript titled “Differential response of the gut microbiome and resistome to antibiotic exposure in infants and adults”, Li et al. set out to measure and explain the differences in adult and infant microbiomes post-antibiotic exposure. To do this, they take advantage of a pair of exceptional study cohorts, 217 adults and 662 infants through the Danish COPSAC organization. The authors use high-throughput sequencing of fecal samples from these cohorts to interrogate their microbiomes for the presence/absence and abundance of antibiotic resistance genes and mobile genetic elements. Within each cohort are numerous samples taken from people with and without antibiotic exposure, this internal control drives much of the authors’ findings. The authors use well established tools and methods to answer their questions, and the statistical treatments seem appropriate, including the use of multiple hypothesis testing corrections.

MAJOR FINDINGS

The most interesting major finding from this manuscript is how cleanly (for a biological measure) *E. coli* presence/absence delineates the antibiotic resistance gene abundance in both cohorts. The authors report that the presence of *E. coli* strains is highly associated with increased resistance gene abundance. While it is known that Enterobacteraceae are often carriers of antibiotic resistance genes, I feel that the striking divide here between + and - *E. coli* presence is a significant finding. The authors also report on differences between infant and adult resistomes, with their major conclusion being that adult resistomes are lower in resistance gene diversity compared to infants, likely due to the immature nature of the infant gut microbiome. Finally, the authors also make a notable contribution by leveraging self-reported antibiotic use in their cohorts to estimate time-to-normalize after antibiotic therapy. The authors report that, again perhaps reflecting their immature and changing nature, infant microbiomes recover more rapidly from antibiotic exposure than adult microbiomes. Interestingly, recovery shows distinct effects reflecting major antibiotic classes. Overall, the authors make a significant contribution to the field. I am often not excited by large sequencing-based studies, but I was pleasantly surprised by the focus and clarity of message in this manuscript.

MAJOR CONCERNS

I do not have any major concerns that would significantly affect the manuscript. I do have some concerns (see below) that, despite being minor, should be addressed.

MINOR CONCERNS

- Line 25 in the abstract (as well as lines 58 and 177 [particularly exaggerated]): The authors make it a point to say how little, if anything, is known about how antibiotic resistance changes with age. This is an exaggeration and should be toned down as there are certainly other papers which work to address this. Without performing a deep dive into the literature (I will leave this to the authors) it appears that two recent papers cover a very similar topic (e.g., PMC8388928 and PMC8550338). A quick glance suggests these other manuscripts contain concordant findings to the one under review, and it would be appropriate to 1) tone down claims about the absence of prior research, and 2) cite appropriate prior research. However, this does not detract from the accomplishments of the manuscript currently under review.
- Line 64: Off target affects on the host body would be interesting to study, especially as it correlates to age (grey baby syndrome?) but I assume here body ought to be replaced by “host microbiome” or something along those lines
- Lines 77, 81, and many other places: The authors frequently refer to their cohorts as 18-year and one-year olds but Table 1 clearly states that these are the median ages and the range, while tight, is not limited to those ages. Please refer to them as median ages in the text.
- Table 1 provides quite a bit of potentially interesting metadata that is not discussed further in the manuscript. For example, would it be worth mentioning potential confounds of greater antibiotic use in infants (touched on a little in the discussion section) or differences in dog ownership (and therefore microbiome diversity) between infants and adults? Also in Table 1, is there a reason why infant family income is not spread across 5 categories like in the adult section?
- The E. coli-led bifurcation of microbiomes is an interesting result with an apparently very strong signal. I want to ask the authors to double check that database bias does not have an influence in this case. As they note, relative abundance of E. coli in the gut microbiome is <2%, could the apparent importance of this taxa, which happens to be one of the most thoroughly sequenced and databased taxa, be a false signal?
- Figure 1ef: I appreciate these figures and how clear they are. e) makes good use of a black bar vs gray bar above the graphs to signal loss of E. coli from the analysis, I’d suggest carrying this over to panel f) as well
- Lines 197-211: Please clarify the biological significance of %ARG variance being un-affected by E. coli removal while ARG relative abundance is highly impacted by this analysis change.
- There is some amount of variability in how antibiotics and antibiotic resistance genes are classified throughout the manuscript. In some cases, the β -lactams are split into penams, cephalosporins, etc and tetracyclines are split into tetracyclines and glycylyclines (fig 1a they are split, but figure 6a they are not?) Please clarify how these splits were made and if they affect the results. If it’s possible, it would be best if category splitting/not-splitting was consistent within the whole manuscript.

Reviewer #2 (Remarks to the Author):

The presented study describes how the gut resistome changes with age, by comparing two different age groups. The results are of significant clinical relevance and some of the results are novel, such as the description of a core resistome found in both age groups, detailed characterisation of mobilome and associated ARGs, and the deduction of longer-lasting effects of antibiotic treatments in adults than in infants. However, other results have been reported by the same authors previously (Li et al, 2022. *Cell Host Microbe*. The infant gut resistome associates with *E. coli*, environmental exposures, gut microbiome maturity, and asthma-associated bacterial composition) such as the bimodal distribution of infant gut resistome driven by *E. coli*.

Comments:

The authors propose that a bimodal distribution of ARGs in the adult gut (Fig 1 & 2) is driven by the *E. coli* species. However, on line 617 (and Supplementary Tables) is a description of another closely related species, *E. flexneri*. The name *E. flexneri* might have been used in the past, but the commonly used name in clinical microbiology (but also in general microbiology) is *Shigella flexneri*. Although *E. coli* and *S. flexneri* are closely related, they represent separate species and are epidemiologically distinct. Recent studies describe the emergence of multidrug-resistant *S. flexneri* clones ([https://doi.org/10.1016/S1473-3099\(22\)00807-6](https://doi.org/10.1016/S1473-3099(22)00807-6), <https://doi.org/10.1016/j.jiph.2017.09.025>) and highlight their pathogenic potential. When having available MAGs for both *E. coli* and *S. flexneri*, what is the reason for classifying ARGs from these two MAGs as only *E. coli*? Combining the MAGs is likely to result in a skewed data interpretation (e.g., lines 100, 469-471).

Similarly, *Shigella dysenteriae* is described as *Escherichia dysenteriae* (Fig. 1h). For the same reason described for *S. flexneri*, *S. dysenteriae* is the officially recognised species name. The authors should correct the naming accordingly.

Please specify in the introduction text that the findings are from a cross-sectional sub-study of the COPSAC cohort.

Abstract – Including the number of infants and adults would be helpful to immediately see the scope of the study.

Figure S1 – Description/definition of “PAM clusters” is missing. Also in Panel b, the numbering of drug classes is not continuous, numbers 11 and 13 are missing. What is the reason for that? And why

the number of drug classes related to MDR ARGs in the lower part of the panel (11) does not match the upper panel (6)?

Lines 92-93 v and 98-99 – There seems to be a repetition in the information of the two sentences, which can be combined in one sentence.

Figure 1 – panel h is mentioned in the text right after panel a, and before panels b, c, d, e, f. The naming of panels should appear consecutively in the text. Can the authors explain why panel d is included? It is not clear why this information is needed.

Fig. S4 describes *E. flexneri*. For reasons described above, the name of *Shigella flexneri* would be preferred. Also, what is the reason of performing the analysis on the genus level? Would there be a gain of a better resolution if the analysis was done at the species level? Similarly for Fig S5 – would species level analysis identify if certain *Bifidobacterium* species are more likely to not carry ARGs than others?

Lines 258-260 Please rewrite the sentence «We also compared the number and type of ARGs or MGEs in those shared ARG-carrying or MGE-carrying bacterial species...» as it is not clear in which species is the comparison is being made. Using the term «shared species» might allude to a scenario where exactly the same species were shared among individuals within the same community/family, which is not likely the case. The y-axis in Fig S7 describes “overlapping species” and that might be a better fitting term.

The term “shared” is used also in other parts of the text (e.g. line 264). I recommend using consistent terminology throughout the text (overlapping or shared), or defining what “shared species” or “overlapping species” mean in the context of the current study.

Figure 3 panel a – what does it mean unclassified? And are these unclassified species or ARGs? Also, would be possible to make the Venn diagrams' size proportional to the numbers they represent?

Line 268 – Please define “unique species” by their abundance. And were these species unique to a few individuals or evenly distributed in infants/ young adults?

Table S4 – as the table is now, it is not possible to see the overlapping (or “shared”) species/ARGs (but also MGE, MDR, Drug classes etc) in the same row. Please recreate the lists that the species/ARGs are ordered alphabetically and correspond across different columns (infant, adult,

“shared”). This would allow to see immediately whether specific species/ARG (MGE, MDR, Drug classes etc) is present in either infants, adults, or both.

Lines 369-372. Is it possible to show the different PAM clusters for adults and infants ARGs in a supplementary table? It would be also interesting to see which ARGs were enhanced by the antibiotic treatment.

Lines 435-438 and Figure 5e – The authors state that “None of the antibiotics had a statistically significant influence on the abundance of the 20 most abundant bacterial species” in the infant gut. However, in Fig 5e the text states “ Members of the 20 most abundant bacterial species whose abundance in the gut differed significantly between ...(bottom) infants who had taken antibiotics ...”. Please correct. Also, the figure does not specify which antibiotic types are shown for infants.

Lines 500- 515. The text and argumentation seem counterintuitive. It is indeed true that facultative anaerobes such a E. coli are the pioneering colonisers of the infant gut. At the same time, E. coli have been shown as the most likely host for a majority of ARGs independent of later disease risks (Lebeaux et al. The infant gut resistome is associated with E. coli and early-life exposures. BMC Microbiol. 2021; 21: 201, Bargheet et al. Development of early life gut resistome and mobilome across gestational ages and microbiota-modifying treatments. eBioMedicine, 2023; 92: 104613). In the context of the presented work (i.e. a cross-sectional study), the sentence “ It is possible that the natural processes of gut maturation may be altered by the presence or abundance of ARGs or ARG-carrying bacteria.” appears very speculative, and the authors should consider toning down such statements in the text.

In the description of quality control steps, the authors should mention the specific command-line options used, such as quality thresholds and the applied trimming parameters. Also, there is no mention of which index was used for alpha diversity and which package was used for the estimation.

Page 721 – Although the data that support the results of the study are available from the corresponding author upon request, for complete transparency and enabling reproducibility of the results, the authors should provide the source data for each figure.

We thank the editors and reviewers for the comments, questions raised and suggestions. We have addressed the suggestions individually below, which have improved the manuscript substantially.

Reviewer #1 (Remarks to the Author):

SUMMARY

In their manuscript titled "Differential response of the gut microbiome and resistome to antibiotic exposure in infants and adults", Li et al. set out to measure and explain the differences in adult and infant microbiomes post-antibiotic exposure. To do this, they take advantage of a pair of exceptional study cohorts, 217 adults and 662 infants through the Danish COPSAC organization. The authors use high-throughput sequencing of fecal samples from these cohorts to interrogate their microbiomes for the presence/absence and abundance of antibiotic resistance genes and mobile genetic elements. Within each cohort are numerous samples taken from people with and without antibiotic exposure, this internal control drives much of the authors' findings. The authors use well established tools and methods to answer their questions, and the statistical treatments seem appropriate, including the use of multiple hypothesis testing corrections.

MAJOR FINDINGS

The most interesting major finding from this manuscript is how cleanly (for a biological measure) *E. coli* presence/absence delineates the antibiotic resistance gene abundance in both cohorts. The authors report that the presence of *E. coli* strains is highly associated with increased resistance gene abundance. While it is known that Enterobacteraceae are often carriers of antibiotic resistance genes, I feel that the striking divide here between + and - *E. coli* presence is a significant finding. The authors also report on differences between infant and adult resistomes, with their major conclusion being that adult resistomes are lower in resistance gene diversity compared to infants, likely due to the immature nature of the infant gut microbiome. Finally, the authors also make a notable contribution by leveraging self-reported antibiotic use in their cohorts to estimate time-to-normalize after antibiotic therapy. The authors report that, again perhaps reflecting their immature and changing nature, infant microbiomes recover more rapidly from antibiotic exposure than adult microbiomes. Interestingly, recovery shows distinct effects reflecting major antibiotic classes. Overall, the authors make a significant contribution to the field. I am often not excited by large sequencing-based studies, but I was pleasantly surprised by the focus and clarity of message in this manuscript.

MAJOR CONCERNS

I do not have any major concerns that would significantly affect the manuscript. I do have some concerns (see below) that, despite being minor, should be addressed.

MINOR CONCERNS

Comment 1 (C1) • Line 25 in the abstract (as well as lines 58 and 177 [particularly exaggerated]): The authors make it a point to say how little, if anything, is known about how antibiotic resistance changes with age. This is an exaggeration and should be toned down as there are certainly other papers which work to address this. Without performing a deep dive into the literature (I will leave this to the authors) it appears that two recent papers cover a very similar topic (e.g., PMC8388928 and PMC8550338). A quick glance suggests these other manuscripts contain concordant findings to the one under review, and it would be appropriate to 1) tone down claims about the absence of prior research, and 2) cite appropriate prior research. However, this does not detract from the accomplishments of the manuscript currently under review.

Reply 1 (R1): We thank the reviewer for the suggestion and agree with the assessment. In the revised paper, we have toned down the statement in line 25 while emphasizing the lack of understanding in current studies of how antibiotics respond to the gut microbiome/resistome across ages. We have rewritten the sentence in line 58, citing relevant references related to ARG profile comparison between infants and adults. Additionally, we have deleted the sentence in line 177.

C2 • Line 64: Off target effects on the host body would be interesting to study, especially as it correlates to age (grey baby syndrome?) but I assume here body ought to be replaced by "host microbiome" or something along those lines

R2: We thank the reviewer for the suggestion. We have revised "host body" into "host microbiome" in the revised paper.

C3 • Lines 77, 81, and many other places: The authors frequently refer to their cohorts as 18-year and one-year olds but Table 1 clearly states that these are the median ages and the range, while tight, is not limited to those ages. Please refer to them as median ages in the text.

R3: We thank the reviewer for the suggestion. We have added a note in rows 77 and 81 in the revised paper, where these two ages first appeared, indicating that these ages are medians.

C4 • Table 1 provides quite a bit of potentially interesting metadata that is not discussed further in the manuscript. For example, would it be worth mentioning potential confounds of greater antibiotic use in infants (touched on a little in the discussion section) or differences in dog ownership (and therefore microbiome diversity) between infants and adults? Also in Table 1, is there a reason why infant family income is not spread across 5 categories like in the adult section?

R4: We agree with the reviewer that including more analysis on the metadata could be interesting. We have conducted the PERMANOVA test to examine the overall effects of environmental factors listed in the table (excluding antibiotics, which have been discussed in detail in the paper) on the adult gut microbiome/resistome. As shown in the table below, these environmental factors did not show significant impacts on ARG composition, while sex had a significant effect on microbial composition, which has been added in the revised paper. In contrast, sex did not have a significant effect on the infant gut microbiome (Adonis test, $P = 0.51$). The effects of environmental factors on the infant gut microbiome and resistome can be found in our previous paper (PMID: 33887206). Furthermore, we conducted a Fisher's test to investigate potential confounding factors associated with increased antibiotic use in infants, including sex, living area, cat/dog ownership, maternal antibiotics during pregnancy, siblings, delivery mode, housing, and family income type. The analysis revealed that infants whose mothers had used antibiotics during pregnancy had a higher likelihood of antibiotic use during their first year of life (Fisher's test, $P = 0.03$, odds ratio = 1.41), and this information has been added to the discussion in the revised paper. As pet ownership has a limited impact on the adult ARG and Microbiome compositions in adults alone analysis ($R^2 < 1\%$ - table below), it is likely not a confounder, however, we have conducted an analysis stratified to children and adults without pets, and the inference in relation to age (infants vs adults) sustains: Full analysis (ARGs | Microbiome: $R^2 = 8.5\% | R^2 = 10\%$, $P < 0.001 | P < 0.001$), stratified analysis (ARGs | Microbiome: $R^2 = 5.2\% | R^2 = 6.0\%$, $P < 0.001 | P < 0.001$). For this reason, we have not included adjusted analysis in this regard. Regarding income, since the infant samples were collected approximately 10 years earlier than the adult samples, and the assessment criteria for household income were not exactly the same for the two cohorts. We have added the detailed household income assessment criteria in the table note of the revised paper.

	Environmental factors	R ²		Adjusted P value	
Adults: ARGs Microbiome	Sex	0.41%	1.2%	0.60	0.0024
	Living area	0.42%	0.62%	0.60	0.34
	Pet	0.59%	0.44%	0.42	0.71
	Siblings	0.40%	0.25%	0.60	0.99
	Smoking	0.44%	0.57%	0.60	0.34
	Alcohol Drink	0.68%	0.38%	0.42	0.83
	Income	0.64%	0.72%	0.42	0.19
	Education	0.70%	0.53%	0.42	0.51

C5 • The E. coli-led bifurcation of microbiomes is an interesting result with an apparently very strong signal. I want to ask the authors the double check that database bias does not have an influence in this case. As they note, relative abundance of E. coli in the gut microbiome is <2%, could the apparent importance of this taxa, which happens to be one of the most thoroughly sequenced and databased taxa, be a false signal?

R5: We are pleased to provide further clarification on this point to the reviewer. Despite E. coli accounting for only approximately 2% of the average abundance in the adult gut, the primary factor driving the differentiation between the two clusters is the difference in E. coli composition. E. coli exhibits the largest difference in relative abundance between the two clusters, with a factor of 66. We agree with the reviewer that E. coli is likely one of the most extensively studied species, and there may be some bias in the database. However, the comparability of the two clusters is based on their reliance on the same database, which means that any biases introduced by the database should theoretically be equal for both clusters. Additionally, we employed RGI software to conduct homology searches for identifying

ARGs against the database. This approach helps reduce database biases and ultimately enabled the identification of 237 species carrying ARGs in infants and 273 species carrying ARGs in adults.

C6 • Figure 1ef: I appreciate these figures and how clear they are. e) makes good use of a black bar vs gray bar above the graphs to signal loss of *E. coli* from the analysis, I'd suggest carrying this over to panel f) as well

R6: We thank the reviewer for the suggestion. We have applied black and gray bars to panel 1f in the revised paper.

C7 • Lines 197-211: Please clarify the biological significance of %ARG variance being un-affected by *E. coli* removal while ARG relative abundance is highly impacted by this analysis change.

R7: We are pleased to clarify this. These results suggest that 1) the difference in *E. coli* composition did not determine the overall variance in ARG profiles in terms of abundance and species between the two cohorts 2) but was the main reason for the differences in gut ARG load between the two cohorts. For the first point, our data analysis suggests that this is mainly due to: 1. large differences in the ARG-carrying bacteria species; 2. differences in the types of ARG carried by these bacteria. These analyses are shown in Fig 2C, 3A and 3B: at least half of the ARG-carrying bacteria were specific to each cohort and about half of the ARGs were also specific to each cohort. The reason for the second point is primarily that the average relative abundance of *E. coli* in the infant gut is much greater than in adults. We assessed ARG load in two ways: the number of ARGs per million genes 2. the relative abundance of ARGs in all genes. We have rephrased the corresponding section to improve the clarity of presenting this point in the revised paper.

C8 • There is some amount of variability in how antibiotics and antibiotic resistance genes are classified throughout the manuscript. In some cases, the β -lactams are split into penams, cephalosporins, etc and tetracyclines are split into tetracyclines and glycylyclines (fig 1a they are split, but figure 6a they are not?) Please clarify how these splits were made and if they affect the results. If it's possible, it would be best if category splitting/not-splitting was consistent within the whole manuscript.

R8: We are pleased to clarify this. The classification of which antibiotics the ARGs are resistant to in Fig. 1a was based on the database's refined classification criteria, to demonstrate in detail the ARG categories detected in the population. In analyzing the antibiotic classes taken by the population in Fig 6a, we combined all antibiotics belonging to the β -lactam class into one big category to obtain more statistical power, to explore the overall impact of a broad range of antibiotic classes on the gut microbiome. Since the research questions explored in the two figures have distinct focal points, we anticipate that the impact on the outcomes should be acceptable.

Reviewer #2 (Remarks to the Author):

The presented study describes how the gut resistome changes with age, by comparing two different age groups. The results are of significant clinical relevance and some of the results are novel, such as the description of a core resistome found in both age groups, detailed characterisation of mobilome and associated ARGs, and the deduction of longer-lasting effects of antibiotic treatments in adults than in infants. However, other results have been reported by the same authors previously (Li et al, 2022. Cell Host Microbe. The infant gut resistome associates with *E. coli*, environmental exposures, gut microbiome maturity, and asthma-associated bacterial composition) such as the bimodal distribution of infant gut resistome driven by *E. coli*.

Comments:

C9: The authors propose that a bimodal distribution of ARGs in the adult gut (Fig 1 & 2) is driven by the *E. coli* species. However, on line 617 (and Supplementary Tables) is a description of another closely related species, *E. flexneri*. The name *E. flexneri* might have been used in the past, but the commonly used name in clinical microbiology (but also in general microbiology) is *Shigella flexneri*. Although *E. coli* and *S. flexneri* are closely related, they represent separate species and are epidemiologically distinct. Recent studies describe the emergence of multidrug-resistant *S. flexneri* clones ([https://doi.org/10.1016/S1473-3099\(22\)00807-6](https://doi.org/10.1016/S1473-3099(22)00807-6), <https://doi.org/10.1016/j.jiph.2017.09.025>) and highlight their pathogenic potential. When having available MAGs for both *E. coli* and *S. flexneri*, what is the reason for classifying ARGs from these two MAGs as only *E. coli*? Combining the MAGs is likely to result in a skewed data interpretation (e.g., lines 100, 469-471).

R9: We are pleased to provide further clarification for the reviewer. The two clusters were primarily driven by differences in *E. coli* composition (Fig. 1g), which was determined based on marker genes using MetaPhlAn. But "*The reads assigned to E. coli by MetaPhlAn were subdivided into two main*

MAGs, one for *E. coli* and the other for *E. flexneri*" based on the GTDB database v202. There were two main reasons for the merger, firstly to accommodate compositional analysis and for concise description, and secondly for comparability with our previous infant results. The latter was based on an older version of GTDB database which did not differentiate between *E. coli* and *E. flexneri*. We agree with the differences between the two as noted by the reviewer, and therefore we further emphasize in the methods that all analyses involving *E. coli* MAGs in Fig. 1 and 2 are a merger of the two, and we have added an illustration of *Shigella flexneri* to the discussion in the revised paper.

C10: Similarly, *Shigella dysenteriae* is described as *Escherichia dysenteriae* (Fig. 1h). For the same reason described for *S. flexneri*, *S. dysenteriae* is the officially recognised species name. The authors should correct the naming accordingly.

R10: We thank the reviewer for the suggestion. We have revised "*Escherichia dysenteriae*" into "*Shigella dysenteriae*" in the revised Fig. 1.

C11: Please specify in the introduction text that the findings are from a cross-sectional sub-study of the COPSAC cohort.

R11: We thank the reviewer for the suggestion. We have added the description in the introduction in the revised paper.

C12: Abstract – Including the number of infants and adults would be helpful to immediately see the scope of the study.

R12: We thank the reviewer for the suggestion. We have added the number of infants and adults in the abstract in the revised paper.

C13: Figure S1 – Description/definition of "PAM clusters" is missing. Also in Panel b, the numbering of drug classes is not continuous, numbers 11 and 13 are missing. What is the reason for that? And why the number of drug classes related to MDR ARGs in the lower part of the panel (11) does not match the upper panel (6)?

R13: We have added a description of "PAM clusters" to the figure legend as suggested. Regarding the missing numbers for MDR ARGs resistant to 11 and 13 drug classes, this was because we did not observe any MDR ARGs with resistance to those specific numbers of drug classes in our study. In the bottom panel of Figure 1b, we displayed a total of 26 drug classes that were found in MDR ARGs. This means that the sum of the unique drug classes in the top panel was also 26. In other words, some drug classes represented by different colors in the top panel overlapped with each other.

C14: Lines 92-93 v and 98-99 – There seems to be a repetition in the information of the two sentences, which can be combined in one sentence.

R14: We thank the reviewer for the suggestion. Indeed, there exists a nuanced contextual distinction between the two statements. The former refers to all the ARGs present, while the latter specifically refers to MDR ARGs within the context.

C15: Figure 1 – panel h is mentioned in the text right after panel a, and before panels b, c, d, e, f. The naming of panels should appear consecutively in the text. Can the authors explain why panel d is included? It is not clear why this information is needed.

R15: We thank the reviewer for the suggestion. We have made adjustments to the order of panels in Fig. 1 and the order of mentions in the context as recommended. Panel d was utilized to illustrate the optimal number of clusters for grouping adults based on ARG composition using PAM clustering analysis. This was done to assess whether it aligns with the bimodal distribution of richness shown in Fig. 1b and the presence of two distinct clusters of ARG abundance depicted in Fig. 1c.

C16: Fig. S4 describes *E. flexneri*. For reasons described above, the name of *Shigella flexneri* would be preferred. Also, what is the reason of performing the analysis on the genus level? Would there be a gain of a better resolution if the analysis was done at the species level? Similarly for Fig S5 – would species level analysis identify if certain *Bifidobacterium* species are more likely to not carry ARGs than others?

R16: We thank the reviewer for the suggestion. We have revised "*E. flexneri*" into "*Shigella flexneri*" in the Fig. S4. Based on the results above, we chose *Escherichia* as the focus of this section of the study. Analyzing at the genus level is our initial attempt to investigate the differences in ARGs between the two populations within the genus *Escherichia*. This preliminary analysis provides a broader perspective for future studies and aligns with our current research focus. In addition,

analyzing multiple species in a genus is computationally challenging. The analysis of Bifidobacterium serves primarily as a comparison to emphasize the higher abundance of ARGs in Escherichia, and we therefore did not extend the analysis.

C17: Lines 258-260 Please rewrite the sentence «We also compared the number and type of ARGs or MGEs in those shared ARG-carrying or MGE-carrying bacterial species...» as it is not clear in which species the comparison is being made. Using the term «shared species» might allude to a scenario where exactly the same species were shared among individuals within the same community/family, which is not likely the case. The y-axis in Fig S7 describes “overlapping species” and that might be a better fitting term. The term “shared” is used also in other parts of the text (e.g. line 264). I recommend using consistent terminology throughout the text (overlapping or shared), or defining what “shared species” or “overlapping species” mean in the context of the current study.

R17: We thank the reviewer for the suggestion. We have made revisions in this section to replace “shared” with “overlapping” in order to avoid potential misunderstandings.

C18: Figure 3 panel a – what does it mean unclassified? And are these unclassified species or ARGs? Also, would be possible to make the Venn diagrams' size proportional to the numbers they represent?

R18: “Unclassified” means that ARG-carrying contigs were not detected in the bins, therefore we cannot track which bacteria those ARGs come from. A description of this has been added to the legend in the revised paper. Regarding the proportional sizing of circles in the Venn diagrams, we have duly considered the reviewer’s suggestion. However, given the presence of substantial numerical disparities, such as the case of ratios like 1:32, within our diagrams, we have opted to retain the current presentation. This choice enables us to provide a balanced and comprehensible visualization, even within the confines of a compact circle.

C19: Line 268 – Please define “unique species” by their abundance. And were these species unique to a few individuals or evenly distributed in infants/ young adults?

R19: In line 268, the term “abundance” refers to the proportion of ARGs present in unique species, representing 6% of the total ARG abundance. We define “unique species” as ARG-carrying species that are exclusively found in either the infant or young adult gut, irrespective of their prevalence or abundance within this study. We have included a description of this definition in the legend of the revised paper.

C20: Table S4 – as the table is now, it is not possible to see the overlapping (or “shared”) species/ARGs (but also MGE, MDR, Drug classes etc) in the same row. Please recreate the lists that the species/ARGs are ordered alphabetically and correspond across different columns (infant, adult, “shared”). This would allow to see immediately whether specific species/ARG (MGE, MDR, Drug classes etc) is present in either infants, adults, or both.

R20: We thank the reviewer for the suggestion. In the revised paper, we have revised Table S4 as follows:

Species carrying ARGs in infant gut	Species carrying ARGs in adult gut	Overlapped Species carrying ARGs in both
51-20 sp001917175	51-20 sp001917175	Yes
Acetatifactor sp900066365		No
Acidaminococcus intestini		No
	Acinetobacter johnsonii	No
Acutalibacter sp000435395		No
Aeromonas caviae		No
Agathobacter faecis	Agathobacter faecis	Yes
Agathobacter rectalis	Agathobacter rectalis	Yes
Agathobacter sp900550845		No

C21: Lines 369-372. Is it possible to show the different PAM clusters for adults and infants ARGs in a supplementary table? It would be also interesting to see which ARGs were enhanced by the antibiotic treatment.

R21: In the revised paper, we have included the ARG clusters for adults and infants in Table S3 as follows. We thank the reviewer for the suggestion. The impact of antibiotic treatment on core ARGs in both cohorts can be found in Fig. S9.

Table S3. The ARG clusters in adult and infant gut.			
ARGs in the adult gut	Clusters	ARGs in the infant gut	Clusters
AAC(3)-lid	Cluster 1 (intermediate-abundance (IA) ARGs)	AAC(3)-lla	Cluster 1 (intermediate-abundance (IA) ARGs)
AAC(6')-lb-cr	Cluster 1 (intermediate-abundance (IA) ARGs)	AAC(6')-le-APH(2'')-la	Cluster 1 (intermediate-abundance (IA) ARGs)
AAC(6')-le-APH(2'')-la	Cluster 1 (intermediate-abundance (IA) ARGs)	AAC(6')-lm	Cluster 1 (intermediate-abundance (IA) ARGs)
AAC(6')-li	Cluster 1 (intermediate-abundance (IA) ARGs)	AAC(6')-ly	Cluster 1 (intermediate-abundance (IA) ARGs)
AAC(6')-lih	Cluster 1 (intermediate-abundance (IA) ARGs)	aadA13	Cluster 1 (intermediate-abundance (IA) ARGs)
AAC(6')-lm	Cluster 1 (intermediate-abundance (IA) ARGs)	acrB	Cluster 2 (differentially abundant(DA) ARGs)

C22: Lines 435-438 and Figure 5e – The authors state that “None of the antibiotics had a statistically significant influence on the abundance of the 20 most abundant bacterial species” in the infant gut. However, in Fig 5e the text states “ Members of the 20 most abundant bacterial species whose abundance in the gut differed significantly between(bottom) infants who had taken antibiotics ...”. Please correct. Also, the figure does not specify which antibiotic types are shown for infants.

R22: We are pleased to clarify more. In the given context, "None of the antibiotics" refers specifically to "None of the three major antibiotics." It implies that none of these three antibiotics had a significant influence on the 20 most abundant bacterial species. As a result, we proceeded to examine the combined effect of the three antibiotics on the infant gut microbiome, as demonstrated in Fig. 6e. To prevent any misunderstandings, we have revised both the context and the figure legend in the revised paper accordingly.

C23: Lines 500- 515. The text and argumentation seem counterintuitive. It is indeed true that facultative anaerobes such as E. coli are the pioneering colonisers of the infant gut. At the same time, E. coli have been shown as the most likely host for a majority of ARGs independent of later disease risks (Lebeaux et al. The infant gut resistome is associated with E. coli and early-life exposures. BMC Microbiol. 2021; 21: 201, Bargheet et al. Development of early life gut resistome and mobilome across gestational ages and microbiota-modifying treatments. eBioMedicine, 2023; 92: 104613). In the context of the presented work (i.e. a cross-sectional study), the sentence “ It is possible that the natural processes of gut maturation may be altered by the presence or abundance of ARGs or ARG-carrying bacteria.” appears very speculative, and the authors should consider toning down such statements in the text.

R23: We appreciate the reviewer's insightful comment regarding the statement on the potential influence of ARGs or ARG-carrying bacteria on natural gut maturation processes. Our viewpoint is rooted in our prior research (PMID: 33887206), where we observed intriguing patterns. Specifically, we found that infants at the age of one, exhibiting elevated levels of E. coli and ARGs, displayed a less mature gut microbiome. In addition, we observed that a microbial composition containing more ARGs was associated with a higher risk of developing asthma at the age of 5. We are grateful to the reviewers for shedding light on this aspect, and we have pointed out the source of the observation in the revised paper. Furthermore, in PMID:33887206, we analyzed more than 2,000 16S sequencing data covering three time points (1 week, 1 month, and 1 year) and observed that the abundance of E. coli in the gut decreased with age, suggesting a negative correlation between E. coli abundance and gut maturation. While we acknowledge the two articles highlighted by the reviewer, which elucidate E. coli's role as a primary host of ARGs, it is noteworthy that the exploration of its association with subsequent disease risk appears to be somewhat limited. To ensure the coherence and precision of our findings' presentation, we have revised the corresponding section in the revised paper.

C24: In the description of quality control steps, the authors should mention the specific command-line options used, such as quality thresholds and the applied trimming parameters. Also, there is no mention of which index was used for alpha diversity and which package was used for the estimation.

R24: We thank the reviewer for the suggestion. In the revised paper, we have added the parameters used with Sickle for quality control. For calculating alpha diversity, we utilized the "observed richness" index for both ARGs and bacterial species. This information can be found in line 683 of the manuscript. The observed richness was determined by counting the number of non-zero ARGs or species present in each sample using a custom code. The R code associated with this manuscript has been uploaded and made available.

C25: Page 721 – Although the data that support the results of the study are available from the corresponding author upon request, for complete transparency and enabling reproducibility of the results, the authors should provide the source data for each figure.

R25: We thank the reviewer for the suggestion. The source data for 93 figures has been uploaded to the public repository and links to the data addresses are provided in the revised paper.

REVIEWERS' COMMENTS

Reviewer #1 (Remarks to the Author):

The authors have satisfactorily answered my questions and responded to my concerns and I have no further comments.

Reviewer #2 (Remarks to the Author):

I appreciate that the authors addressed all the comments to a satisfactory degree. Besides an error on line 462, I have no further comments.

Line 462: Kindly correct to "the most prevalent Bifidobacterium species in the adult gut".

Reviewer #1 (Remarks to the Author):

The authors have satisfactorily answered my questions and responded to my concerns and I have no further comments.

Reviewer #2 (Remarks to the Author):

I appreciate that the authors addressed all the comments to a satisfactory degree. Besides an error on line 462, I have no further comments.

Comment: Line 462: Kindly correct to "the most prevalent Bifidobacterium species in the adult gut".

Reply: We thank the reviewer for the suggestion. We have corrected the sentence to "the most prevalent Bifidobacterium species in the adult gut" in the revised paper.